# The mycobacterial phosphatase PtpA regulates the expression of host genes and promotes cell proliferation

Jing Wang[1], Pupu Ge[1,2], Lihua Qiang[1,3], Feng Tian[4], Dongdong Zhao[1,2], Qiyao Chai[1,2], Mingzhao Zhu [5], Rongbin Zhou[6], Guangxun Meng[7], Yoichiro Iwakura [8], George Fu Gao[1,2] & Cui Hua Liu[1,2]

*Mycobacterium tuberculosis* PtpA is a secreted effector protein that dephosphorylates several proteins in the host cell cytoplasm, such as p-JNK, p-p38, and p-VPS33B, leading to suppression of host innate immunity. Here we show that, in addition, PtpA enters the nucleus of host cells and regulates the expression of host genes, some of which are known to be involved in host innate immunity or in cell proliferation and migration (such as *GADD45A*). PtpA can bind directly to the promoter region of *GADD45A* in vitro. Both phosphatase activity and DNA-binding ability of PtpA are important in suppressing host innate immune responses. Furthermore, PtpA-expressing *Mycobacterium bovis* BCG promotes proliferation and migration of human lung adenoma A549 cells in vitro and in a mouse xenograft model. Further research is needed to test whether mycobacteria, via PtpA, might affect cell proliferation or migration in humans.

[1] CAS Key Laboratory of Pathogenic Microbiology and Immunology, Institute of Microbiology, Chinese Academy of Sciences, Beijing 100101, China. [2] Savaid Medical School, University of Chinese Academy of Sciences, Beijing 101408, China. [3] Institute of Health Sciences, Anhui University, Hefei 230601, China. [4] Center for Bioinformatics, Peking University, Beijing 100871, China. [5] CAS Key Laboratory of Infection and Immunity, Institute of Biophysics, Chinese Academy of Sciences, Beijing 100101, China. [6] Institute of Immunology and the CAS Key Laboratory of Innate Immunity and Chronic Disease, Chinese Academy of Sciences Center for Excellence in Molecular Cell Sciences, School of Life Sciences and Medical Center, University of Science and Technology of China, Hefei 230027, China. [7] CAS Key Laboratory of Molecular Virology and Immunology, Institut Pasteur of Shanghai, Chinese Academy of Sciences, Shanghai 200031, China. [8] Division of Experimental Animal Immunology, Center for Animal Disease Models, Research Institute for Biomedical Sciences, Tokyo University of Science, Chiba 278-0022, Japan. Jing Wang and Pupu Ge contributed equally to this work. Correspondence and requests for materials should be addressed to C.H.L. (email: liucuihua@im.ac.cn)

Tuberculosis (TB) remains a leading cause of death and disability worldwide. According to data from the World Health Organization (WHO)[1], ~10.4 million people were estimated to have fallen ill with TB and 1.4 million people died from TB in 2015. *Mycobacterium tuberculosis* (Mtb), the etiological agent of the disease, survives inside the host macrophages either in an active or non-replicative state. The treatment of active TB requires at least 6 months, which often leads to the emergence

**Fig. 1** PtpA is present both in the cytoplasm and nucleus of host cells. **a** Confocal microscopy of U937 cells infected with WT BCG, or BCG ΔPtpA, or BCG (ΔPtpA + PtpA), or BCG (ΔPtpA + D126A) strain. Cells were infected at a multiplicity of infection (MOI) of 10 for 6 h. PtpA and PtpB were detected using primary antibody specific for PtpA or PtpB followed by DyLight 488-conjugated goat anti-rabbit IgG secondary antibody (*green*). Bacteria were stained with pHrodo Red succinimidyl (NHS) ester (*Red*). Nuclei were stained with DNA-binding dye (*DAPI*) (*blue*). Scale bars, 10 μm. *Right*, the percentage of cells with nuclear localization of PtpA or PtpB (a total of 200 cells were counted) and the survival of BCG strains in U937 cells infected for 6 h. **b** Time course analysis for intracellular localization of PtpA at early time points post-infection (2–6 h). Cells were stained as in **a**. *Right*, the percentage of cells with nuclear localization of PtpA and the survival of BCG strains in U937 cells infected for 2–6 h. **c** Immunoblot analysis of PtpA in U937 cells stably expressing Myc-tagged Mtb PtpA. Cells were collected for fractionation to obtain the cytosolic fraction and the pellets containing the nuclei for immunoblot analysis. α-Tubulin and PARP were used as markers for the cytosolic and nuclear fractions, respectively. **d** Immunoblot analysis of PtpA and PtpB in U937 cells infected as in **a**. Cells were collected for fractionation to obtain the cytosolic fraction and the pellets containing the nuclei for immunoblot analysis as in **c**. *P < 0.05 and **P < 0.01 (unpaired two-tailed Student's *t*-test). Data are representative of one experiment with at least three independent biological replicates (**a**, **b**; mean and s.e.m., n = 3). Full blots are shown in Supplementary Fig. 10

of multidrug-resistant Mtb strains due to inadequate treatment or poor patient compliance. WHO reported that about half of the patients with multidrug-resistant TB are not successfully treated, and the emergence of drug-resistant TB has become a major global threat[1–4]. Thus, it is urgent for us to better understand the molecular mechanisms of the interactions between Mtb and host immune system in order to identify new effective therapeutic targets.

Mtb PtpA is a secreted, low-molecular-weight protein tyrosine phosphatase (PTP) that is important for Mtb pathogenicity in vivo but not essential for Mtb growth in vitro[5]. The crystal structure of Mtb PtpA revealed the PTP loop (residues 11–18) in its active site, along with three conserved active-site residues including Cys11, Arg17, and Asp126. Mutations of those three residues (C11A, R17A, and D126A) in Mtb PtpA cause loss of its phosphatase activity[6]. Mtb PtpA can prevent phagosome-lysosome fusion by dephosphorylating host protein VPS33B, and prevent phagosome acidification by binding to subunit H of the macrophage V-ATPase complex to block V-ATPase trafficking[7, 8]. Furthermore, binding of Mtb PtpA to ubiquitin via a ubiquitin-interacting motif-like region activates PtpA to dephosphorylate JNK, p38, and VPS33B, leading to suppression of innate immunity. Mtb PtpA can also suppress the activation of NF-κB by competitively binding to the Npl4 zinc-finger domain of TAB3 independently of its phosphatase activity[9].

Those previous studies were mainly focused on the regulatory function of Mtb PtpA in the cytoplasm of host cells. Here, we show that Mtb PtpA is not only present in the cytoplasm but also in the nucleus of host cells. Using chromatin immunoprecipitation followed by sequencing (ChIP-seq) analysis[10, 11], we find that nuclear PtpA interacts with host DNA. PtpA appears to regulate the transcription of a variety of protein-coding genes, some of which are known to be involved in host innate immune signaling, cell proliferation, and migration. In addition, PtpA-expressing *Mycobacterium bovis* Bacillus Calmette-Guerin (BCG) promotes cell proliferation and migration of a human lung adenoma cell line in vitro and in a mouse xenograft model. Our findings reveal additional mechanisms by which Mtb PtpA inhibits host innate immunity. Further research is needed to test whether mycobacteria, via PtpA, might affect cell proliferation or migration in humans.

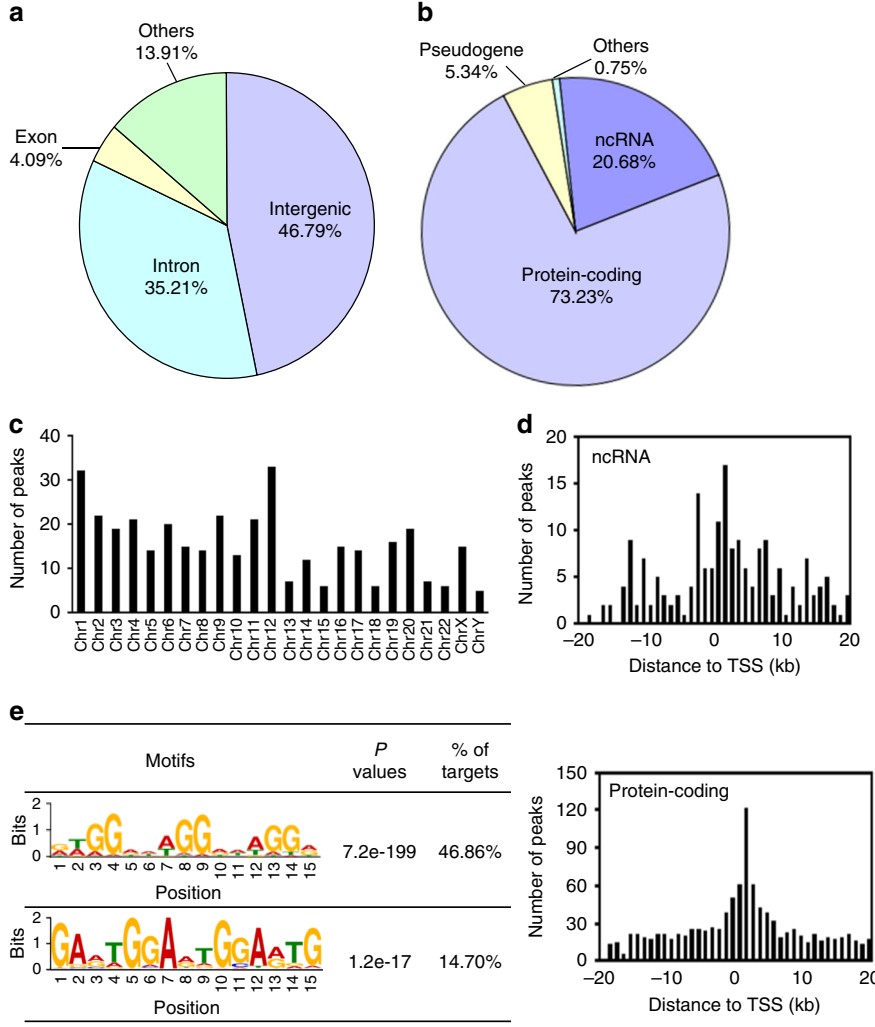

**Fig. 2** Genome-wide analysis of potential PtpA-binding sites in U937 cells. **a** Genomic distribution of potential PtpA-binding regions in U937 cells. **b** Biotype distribution of potential PtpA-binding regions in U937 cells. **c** Distribution of the chromosomal location of PtpA-specific ChIP-seq signals (fold-enrichment > 10). **d** The number and position of the potential PtpA-binding sites along 20 kb from the nearest TSS of the ncRNAs or protein-coding genes, respectively. **e** PtpA consensus motifs defined by MEME motif analysis of the sequences of PtpA ChIP-seq peaks. *P* values were calculated by TOMTOM match statistics. "% of targets" represents the percentage of motif-containing fractions in potential PtpA targets. The size of each letter was proportional to the frequency of each nucleotide in that position within the consensus motif

## Results

**Identification of PtpA in the nucleus of host cells.** Previous studies on the regulatory function of Mtb PtpA were mainly focused on how it interferes with the innate immune system as a phosphatase in the cytoplasm. The amino acid sequence of PtpA from *Mycobacterium bovis* BCG is identical to that of Mtb PtpA. With an aim to probe the subcellular location of PtpA in host cells, we performed confocal microscopy experiments using different BCG strains including wild-type (WT) BCG, PtpA-deleted BCG (BCG ΔPtpA), BCG ΔPtpA complemented with WT PtpA (ΔPtpA + PtpA) and BCG ΔPtpA complemented with phosphatase-inactive PtpA (ΔPtpA + D126A). Our results indicated that some PtpA, but not PtpB (another mycobacterial secreted PTP[12]), entered the nucleus of human macrophage-like U937 cells (which were differentiated from U937 human monocytic cells) during BCG infection, and that this nuclear localization was independent of the phosphatase activity of PtpA (Fig. 1a). We also conducted a time course analysis for intracellular localization of PtpA, and observed that the nuclear localization of PtpA increased over time within 6 h post-infection

(Fig. 1b). Consistently, overexpressed PtpA was also largely co-localized with the nucleus of U937 cells and human lung adenoma A549 cells (Supplementary Fig. 1a–c). We then further performed immunoblot analysis of PtpA in U937 cells stably expressing Myc-tagged Mtb PtpA. PtpA was not only present in the cytoplasm but also in the nucleus of U937 cells (Fig. 1c). Consistent results were also obtained from the immunoblot analysis of U937 cells infected with different BCG strains including WT BCG, BCG ΔPtpA, BCG (ΔPtpA + PtpA), and BCG (ΔPtpA + D126A) (Fig. 1d). Thus, PtpA is present both in the cytoplasm and the nucleus of host cells during BCG infection.

**Genome-wide ChIP-seq analysis of potential PtpA-binding sites.** To gain new insights into how PtpA regulates host cellular processes during mycobacterial infection, ChIP-seq analysis using Flag-tagged specific antibody was conducted in U937 cells. By comparing with the "Input DNA", a total of 3351 potential PtpA-binding sites were detected by ChIP-seq (Supplementary Data 1), of which 46.79% (1568) were localized in intergenic regions and

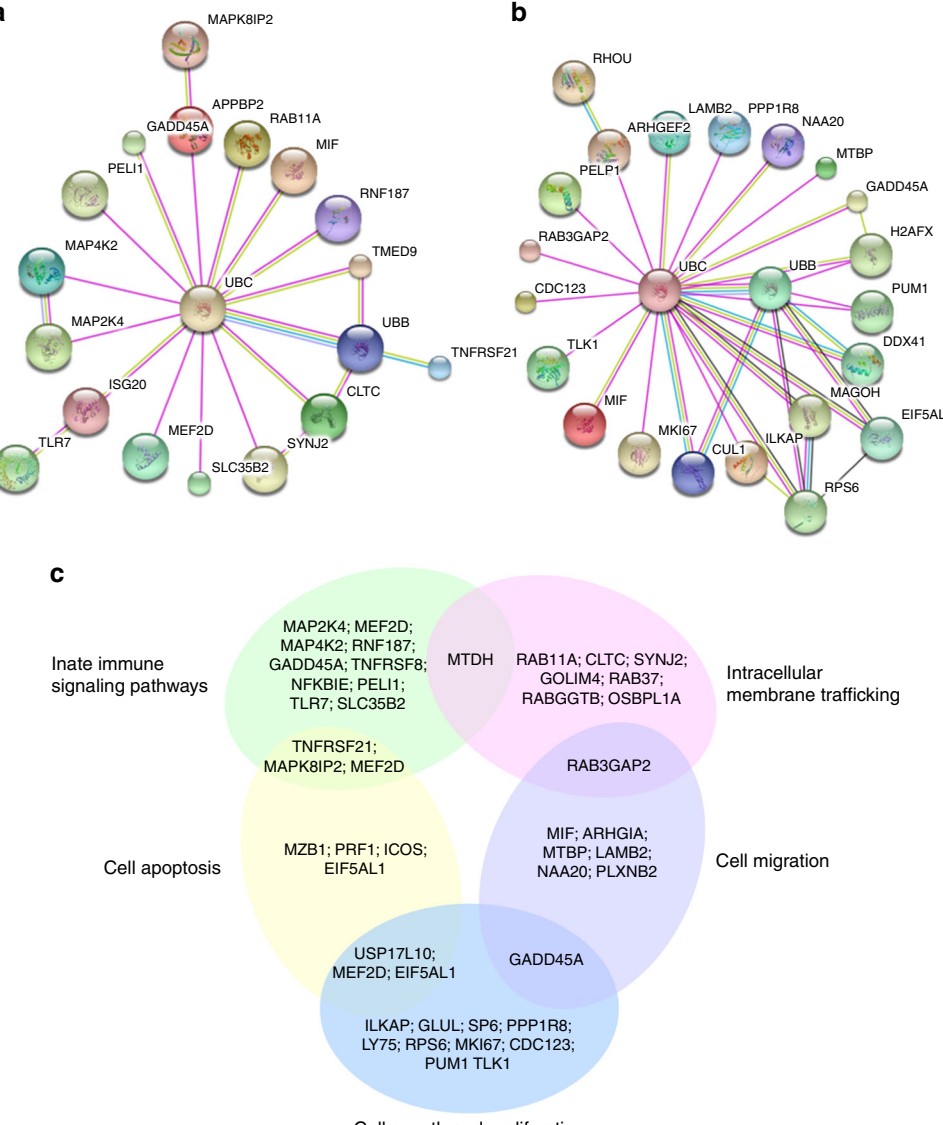

**Fig. 3** Network analysis of proteins potentially regulated by PtpA. **a** A subnet of candidate PtpA-regulated proteins involved in the signaling pathways of the innate immune system. **b** A subnet of candidate PtpA-regulated proteins involved in cell proliferation and migration. **c** Classification of candidate PtpA-regulated proteins, whose gene promoter regions (±2 kb from the TSS) were potentially regulated by PtpA

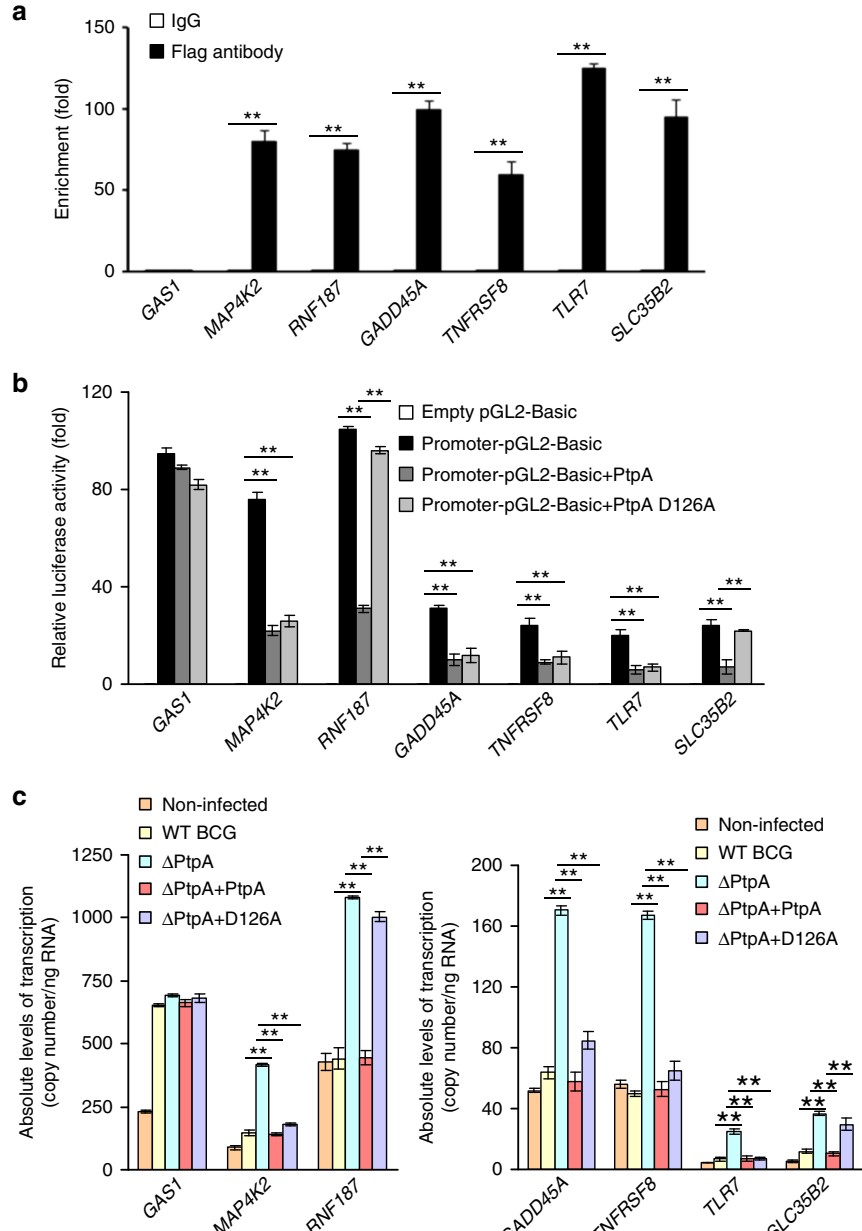

**Fig. 4** Regulation of potential target gene transcription by PtpA. **a** ChIP-qPCR assay for PtpA enrichment in the promoters of potential target genes in U937 cells. ChIP assay was conducted using Flag-tag rabbit antibody followed by qPCR in U937 cells stably expressing PtpA. Enrichment was calculated relative to normal rabbit IgG control. **b** Luciferase reporter assay of potential target gene transcription-regulating activities of PtpA in HEK293T cells. Promoter-cloning vector (promoter-pGL2-Basic) was co-transfected into HEK293T cells with 1 μg of empty vector or vectors encoding full-length WT PtpA or PtpA D126A mutant. The empty pGL2-Basic vector was used as negative control. Resultant luciferase activities were expressed as relative luciferase activities normalized to the pRL-TK activity. **c** QRT-PCR analysis of the transcriptional regulation of potential target genes by PtpA. *GAS1*, *MAP4K2*, *RNF187*, *GADD45A*, *TNFRSF8*, *TLR7*, and *SLC35B2* mRNAs levels were determined in U937 cells infected with WT BCG, BCG ΔPtpA, BCG (ΔPtpA + PtpA), or BCG (ΔPtpA + D126A) for 12 h. The copy numbers of mRNA were calculated based on the *Gapdh* standard curve. *$P < 0.05$ and **$P < 0.01$ (two-tailed unpaired *t*-test). Data are representative of one experiment with at least three independent biological replicates (**a**–**c**; mean and s.e.m., $n = 3$)

35.21% (1180) were localized in intragenic regions (Fig. 2a). Among the 3351 potential PtpA-binding sites analyzed, 73.23% (2454) were identified as protein-coding associated regions, while the remaining ones were classified as different types of noncoding RNAs (ncRNAs) (20.68%, 693) and pseudogenes (5.34%, 179) (Fig. 2b). We then analyzed the distribution of those 3351 potential PtpA-binding regions on chromosomes, and found that they were highly enriched in chromosomes 1 and 12 (Fig. 2c and Supplementary Fig. 2). Further analysis of the PtpA-binding peaks showed that the majority of them were present within 5 kb from the transcription start site (TSS) of nearest protein-coding genes or ncRNAs (Fig. 2d). Two PtpA consensus motifs were identified by MEME motif analysis of the DNA sequences enriched in potential Mtb PtpA-binding regions (Fig. 2e)[13]. Together, these results suggest that in addition to its regulatory function in the cytoplasm of host cells, PtpA may also play a role in regulating gene transcription by potentially binding (directly or indirectly) to gene promoter regions in the nucleus of host cells.

**Network analysis of the candidate PtpA-regulated proteins**. We then focused our attention on the protein-coding associated regions potentially bound by PtpA. Among 2454 such regions, 280 (11.41%) occurred in the promoter region (±2 kb from the TSS) of known RefSeq genes, suggesting a potential transcription regulatory function of PtpA for those genes (Supplementary Data 2). We then conducted a protein–protein interaction network analysis for proteins encoded by those 280 potential PtpA target genes using STRING 10.0[14], a bioinformatic tool that maps protein–protein associations based on evidence channels (Supplementary Fig. 3). Gene Ontology (GO) enrichment analysis[15] was also performed to classify the functions of those 280 proteins (Supplementary Fig. 4). Further analysis showed that a large number of candidate PtpA-regulated proteins (such as TNFRSF8

and MAP4K2)[16, 17] are involved in the regulation of innate immune signaling pathways (Fig. 3a), which was consistent with the previously reported function of PtpA[9]. Also, many of the proteins are associated with cell proliferation and migration (such as GADD45A)[18] (Fig. 3b). Other proteins are associated with intracellular membrane trafficking (such as RAB11A)[19] and cell apoptosis (such as TNFRSF21)[20] (Fig. 3c). These results suggested that, in addition to its immuno-regulatory role, PtpA might play a potential role in host cell proliferation and migration.

**Regulatory effects of PtpA on potential target gene transcription.** In order to verify the ChIP-seq data, seven potential PtpA target genes known to play roles in regulating host innate

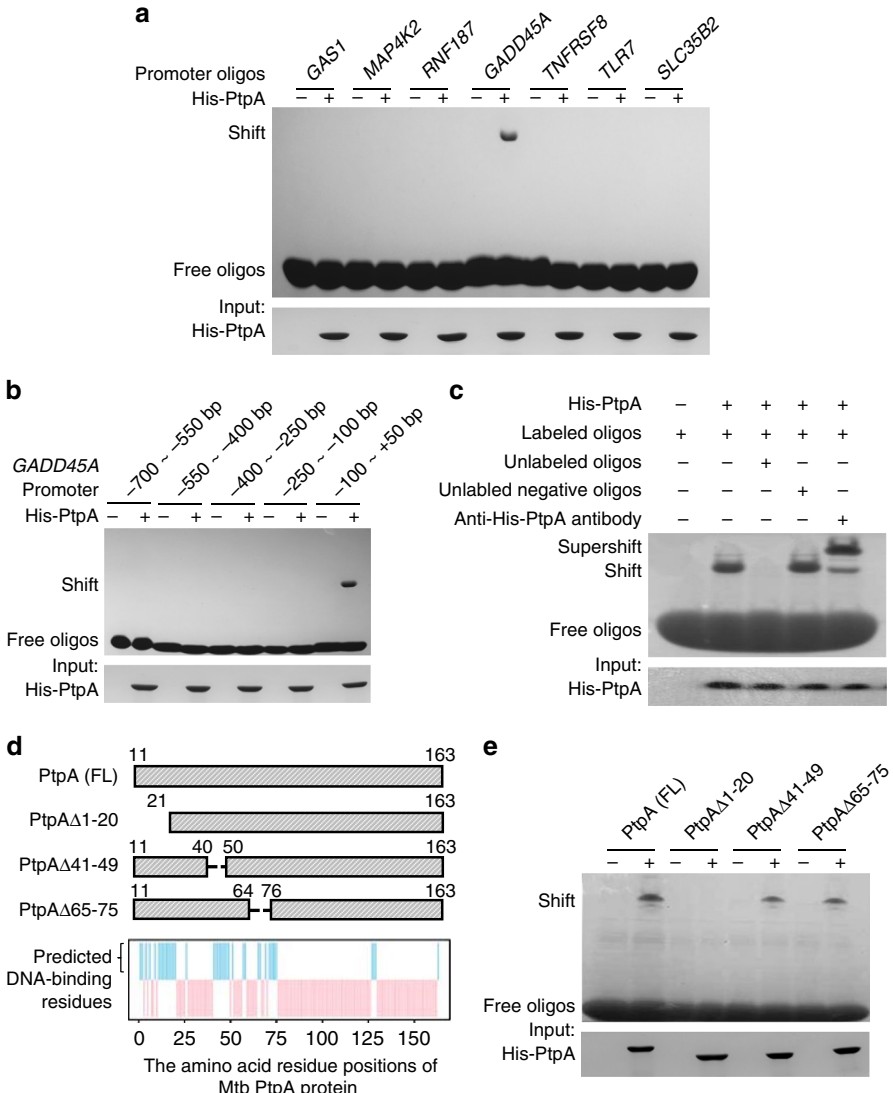

**Fig. 5** Identification of the DNA-binding region of potential target genes by PtpA. **a** EMSA analysis of interactions between His-tagged Mtb PtpA and its potential targeting gene promoters. The promoters of seven genes (10 nM) were amplified and incubated with 10 μg of purified His-PtpA. **b** EMSA analysis of interactions between His-tagged PtpA and five promoter fragments of *GADD45A*. The promoter fragments of *GADD45A* (10 nM) were amplified and incubated with 10 μg of purified His-PtpA. **c** Competitive EMSA analysis of specific interactions between PtpA and the promoter fragment of *GADD45A* (−100 bp to + 50 bp). Biotinylated oligos (−100 bp to + 50 bp) was used to determine their interactions with purified 10 μg His-tagged PtpA protein. Unlabeled positive (−100 bp to + 50 bp) and negative oligos (−250 bp ~ −100 bp) were added in lane 3 or 4 as specific competitor sequences. Lane 5 displayed a super-shift in the presence of anti-His-PtpA antibody. **d** The amino acid residue positions of Mtb PtpA potentially involved in interactions with DNA were predicted by DP-bind (*blue bars*). *Top*, schematic representation of PtpA deletion constructs used for EMSA analysis in **e**. *FL*, full-length. **e** EMSA analysis of interactions between Mtb PtpA proteins and its targeting promoter fragment (−100 bp to + 50 bp from the TSS of *GADD45A*). The biotinylated oligos were incubated with 10 μg of purified full-length or truncated Mtb PtpA proteins. Full blots are shown in Supplementary Fig. 10

immunity as well as cell proliferation and migration, were selected for ChIP-qPCR assay: *GAS1*, *MAP4K2*, *RNF187*, *GADD45A*, *TNFRSF8*, *TLR7*, and *SLC35B2*. The results showed that PtpA was highly enriched in the promoters of *MAP4K2*, *RNF187*, *GADD45A*, *TNFRSF8*, *TLR7*, and *SLC35B2*, but not *GAS1*, in U937 cells, suggesting that except for *GAS1*, the promoters of other six examined genes could be direct or indirect PtpA targets (Fig. 4a). To further test the regulatory roles of PtpA, we inserted the promoter regions of the target genes into the pGL2-Basic vector to create a series of reporter plasmids and performed luciferase reporter assays in U937 cells co-transfected with those reporter plasmids as well as plasmids encoding WT PtpA or PtpA D126A. We found that WT PtpA suppressed the transcription activities induced by the promoters of the PtpA-targeted genes. Interestingly, the phosphatase-inactive variant PtpA D126A suppressed the transcription levels of *MAP4K2*,

*TNFRSF8*, *GADD45A*, and *TLR7*, but not those of *RNF187* and *SLC35B2*. These results suggest that PtpA might regulate the transcription of its target genes either in a phosphatase activity-dependent (*RNF187* and *SLC35B2*) or phosphatase activity-independent manner (*MAP4K2*, *TNFRSF8*, *GADD45A*, and *TLR7*) (Fig. 4b). We then performed quantitative real-time PCR (qRT-PCR) in U937 macrophages. We found that the transcription of the target genes was elevated in cells infected with BCG ΔPtpA strain at 12 h post-infection, but was suppressed in cells infected with WT BCG or BCG (ΔPtpA + PtpA) strain. Consistent with the data from luciferase report assays, the BCG (ΔPtpA + D126A) strain suppressed the transcription levels of *MAP4K2*, *TNFRSF8*, *GADD45A*, and *TLR7*, but not those of *RNF187* and *SLC35B2* (Fig. 4c). qRT-PCR analysis of the transcription levels of PtpA-targeted genes in U937 macrophages infected with *Mycobacterium smegmatis* strains including WT-*M.*

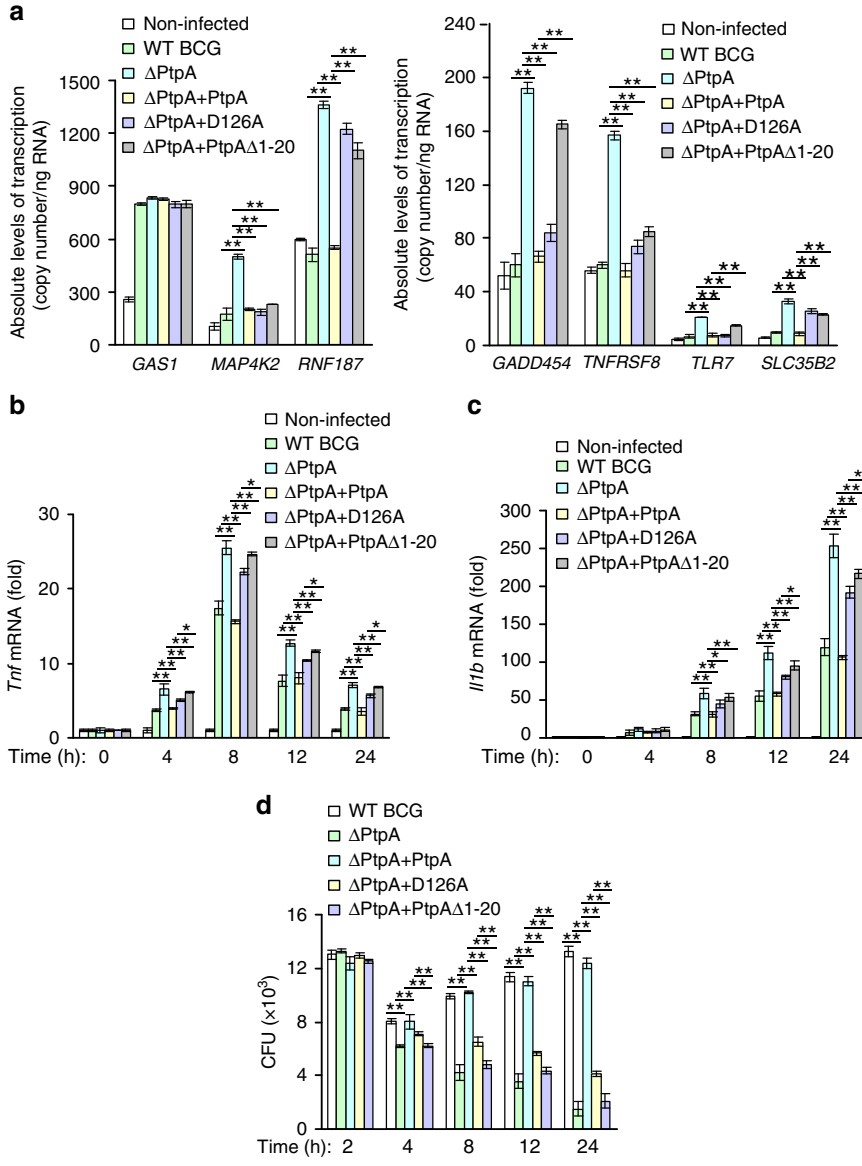

**Fig. 6** The DNA-binding region of PtpA contributes to the inhibition of immune responses. **a** Quantitative PCR analysis of the transcriptional regulation of seven potential target genes by PtpA. *GAS1*, *MAP4K2*, *RNF187*, *GADD45A*, *TNFRSF8*, *TLR7*, and *SLC35B2* mRNAs levels were determined in U937 cells infected with WT BCG, BCG ΔPtpA, BCG (ΔPtpA + PtpA), BCG (ΔPtpA + D126A), or BCG (ΔPtpA + PtpA Δ1-20) at a MOI of 10 for 12 h. mRNA copy numbers were calculated based on the *Gapdh* standard curve. **b**, **c** Quantitative PCR analysis of *Tnf* mRNA **b** and *Il1b* mRNA **c** in U937 cells. Cells were infected for 1-24 h as in **a**. **d** Survival of BCG strains in U937 cells treated as in **b**, **c**. *$P < 0.05$ and **$P < 0.01$ (unpaired two-tailed Student's $t$-test). Data are representative of one experiment with at least three independent biological replicates (**a**–**d**; mean and s.e.m., $n = 3$)

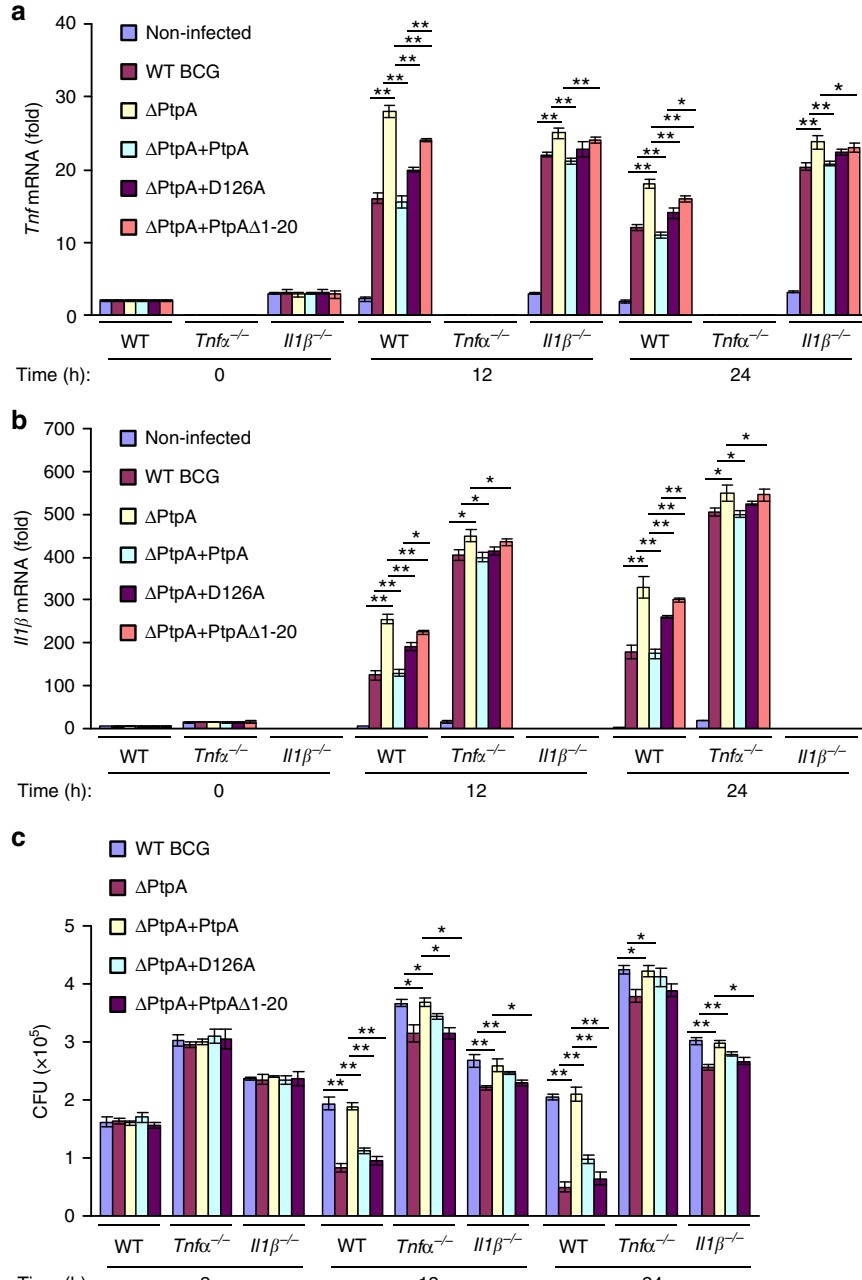

**Fig. 7** Immune suppression by nuclear PtpA during mycobacterial infection is partially dependent on TNF and IL1β. **a, b** Quantitative PCR analysis of *Tnf* mRNA **a** and *Il1b* mRNA **b** in bone marrow derived macrophages (*BMDMs*) from WT, or *Tnfα*$^{-/-}$, or *Il1β*$^{-/-}$ mice. Cells were infected with WT BCG, BCG ΔPtpA, BCG (ΔPtpA + PtpA), BCG (ΔPtpA + D126A), or BCG (ΔPtpA + PtpA Δ1-20) at a MOI of 10 for 0–24 h. Results are presented relative to those of the control gene *Gapdh*. **c** Survival of mycobacteria in WT, or *Tnfα*$^{-/-}$, or *Il1β*$^{-/-}$ BMDMs treated as in **a**. *$P < 0.05$ and **$P < 0.01$ (unpaired two-tailed Student's *t*-test). Data are representative of one experiment with at least three independent biological replicates (**a–c**; mean and s.e.m., $n = 3$)

smegmatis, PtpA-overexpressing *M. smegmatis* (PtpA-*M. smegmatis*), and PtpA D126A-overexpressing *M. smegmatis* (PtpA D126A-*M. smegmatis*) led to the same conclusions as the experiments conducted with BCG strains (Supplementary Fig. 5a–g). Taken together, our results suggest that PtpA may regulate the transcription of certain host genes involved in host innate immunity, as well as other genes involved in cell proliferation and migration, during mycobacterial infection, either in a phosphatase-dependent or -independent manner.

**PtpA directly binds to the promoter region of *GADD45A*.** To determine whether PtpA directly binds to and regulates the

transcription of target genes, we amplified the promoter regions (−700 bp to + 50 bp) of seven potential PtpA target genes (*GAS1*, *RNF187*, *GADD45A*, *TNFRSF8*, *TLR7*, *MAP4K2*, and *SLC35B2*) and performed electrophoretic mobility shift assay (EMSA) with His-tagged PtpA. As shown in Fig. 5a, only the *GADD45A* promoter region could be bound by PtpA directly. We then constructed five short DNA fragments within the *GADD45A* promoter region and repeated the EMSA analysis. We found that the PtpA-binding site was located on the region ranging from −100 bp upstream to + 50 bp downstream of the TSS (Fig. 5b). We then further confirmed the specificity of the binding between PtpA and its putative target *GADD45A* promoter fragment (−100

bp to + 50 bp) by EMSA. As shown in Fig. 5c, the biotinylated *GADD45A* promoter fragment (−100 bp to + 50 bp) formed specific complex with purified His-tagged PtpA and displayed a super-shift in the presence of anti-His-PtpA antibody. Unlabeled competitor *GADD45A* promoter fragment (−100 bp to + 50 bp), but not an unspecific DNA fragment (*GADD45A* promoter region from −250 bp upstream to −100 bp downstream of the TSS), out-competed the labeled fragment for binding to PtpA.

To further determine the DNA-binding region within PtpA, we predicted residue positions of PtpA involved in interactions with DNA by DP-bind (Fig. 5d), a web server for sequence-based prediction of DNA-binding residues in DNA-binding proteins[21], and constructed three deletion mutants of PtpA (PtpA Δ1-20, PtpA Δ41-49, and PtpA Δ65-75) to test their in vitro interactions with the *GADD45A* promoter fragment (−100 bp to + 50 bp) by EMSA analysis. As shown in Fig. 5e, the deletion of the N-

terminal 1–20 amino acids, but not other regions of PtpA, abolished their DNA-binding activities. These results indicate that the N-terminal region (amino acids 1–20) of PtpA may participate in binding to its target DNA.

**Nuclear PtpA suppresses host immune responses to BCG.** We then examined the regulatory role of nuclear PtpA on host innate immune responses during BCG infection. We examined the transcription levels of PtpA-targeting genes in U937 macrophages infected with BCG strains at 12 h post-infection, and found that the transcription levels of *RNF187, GADD45A, TLR7*, and *SLC35B2*, but not those of *MAP4K2* and *TNFRSF8*, were elevated in cells infected with the PtpA Δ1-20-complemented strain BCG (ΔPtpA + PtpA Δ1-20), which had a truncated PtpA losing its DNA-binding region, in comparison with that of WT BCG strain

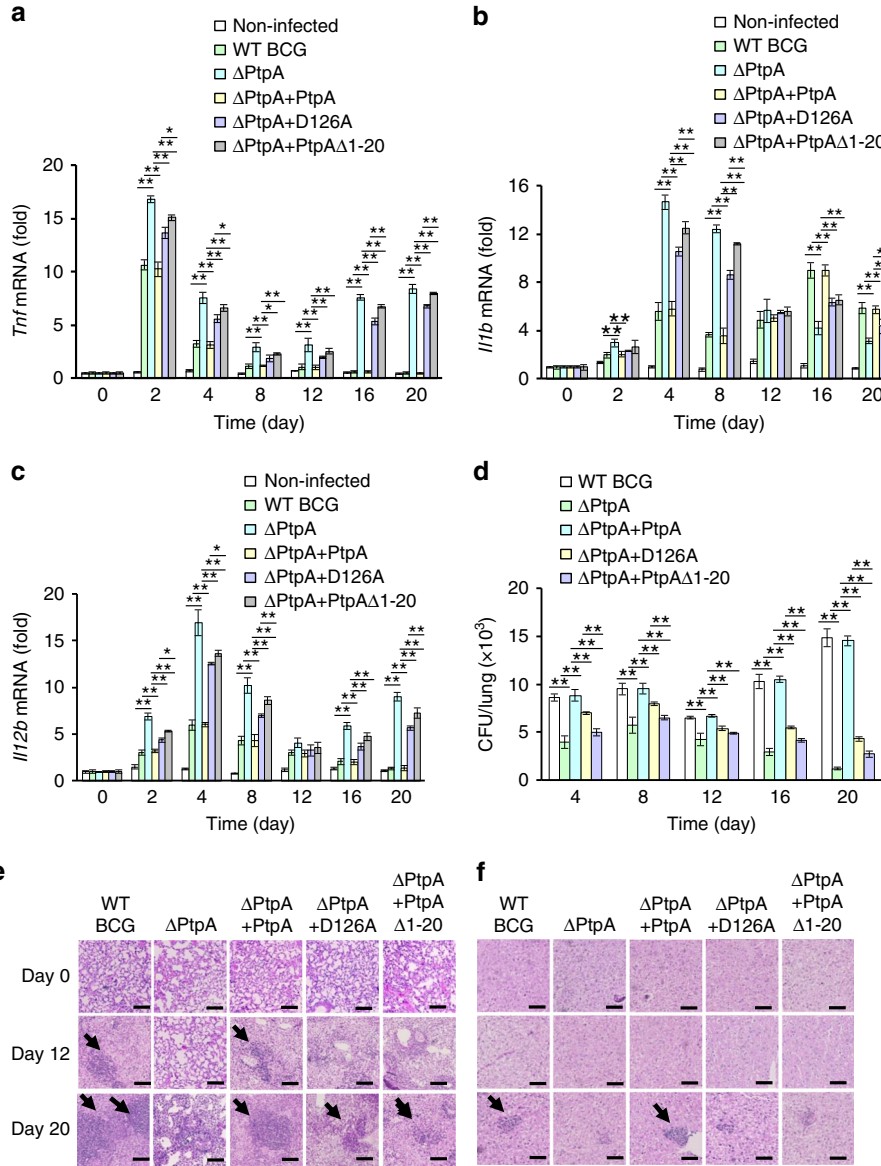

**Fig. 8** Phosphatase activity and DNA-binding ability of PtpA contribute to innate immune suppression during infection in vivo. **a–c** Quantitative PCR analysis of *Tnf* mRNA **a**, *Il1b* mRNA **b** and *Il12b* mRNA **c** in splenic cells from mice. BALB/c nude mice were infected intratracheally with 2 × 10⁶ of WT BCG, BCG ΔPtpA, BCG (ΔPtpA + PtpA), BCG (ΔPtpA + D126A), or BCG (ΔPtpA + PtpA Δ1-20) strain for 0–20 days. **d** Bacterial load in homogenates of lungs from mice treated as in **a–c**. **e, f** Hematoxylin and eosin (H&E) staining of lungs **e** and livers **f** of mice treated as in **a–c**. *Arrows* indicate foci of cellular infiltration. *Scale bars*, 200 µm. *P < 0.05 and **P < 0.01 (unpaired two-tailed Student's *t*-test). Data are representative of one experiment with two independent biological replicates (**a–d**; mean and s.e.m. of n = 6 mice per group)

(Fig. 6a). It should be noted that the deletion of the DNA-binding region (which contains the N-terminal 20 amino acids) of PtpA also abolished its phosphatase activity as revealed previously[6] and confirmed as well in this study (Supplementary Fig. 6). Thus, the N-terminal DNA-binding region of PtpA might also contribute to its role in innate immune suppression since the phosphatase activity of PtpA has been demonstrated to be critical for such regulatory function of PtpA[9]. We next investigated the expression

of inflammatory cytokines, such as tumor necrosis factor (TNF) and IL-1β, as well as the intracellular survival of BCG in U937 cells infected with WT BCG, BCG ΔPtpA, BCG (ΔPtpA + PtpA), BCG (ΔPtpA + D126A), or BCG (ΔPtpA + PtpA Δ1-20). As expected, we found that deletion of PtpA promoted the production of TNF and IL-1β and decreased bacterial survival in U937 cells, and the truncated PtpA (PtpA Δ1-20) abolished the PtpA-mediated suppression of host cytokine production and promotion

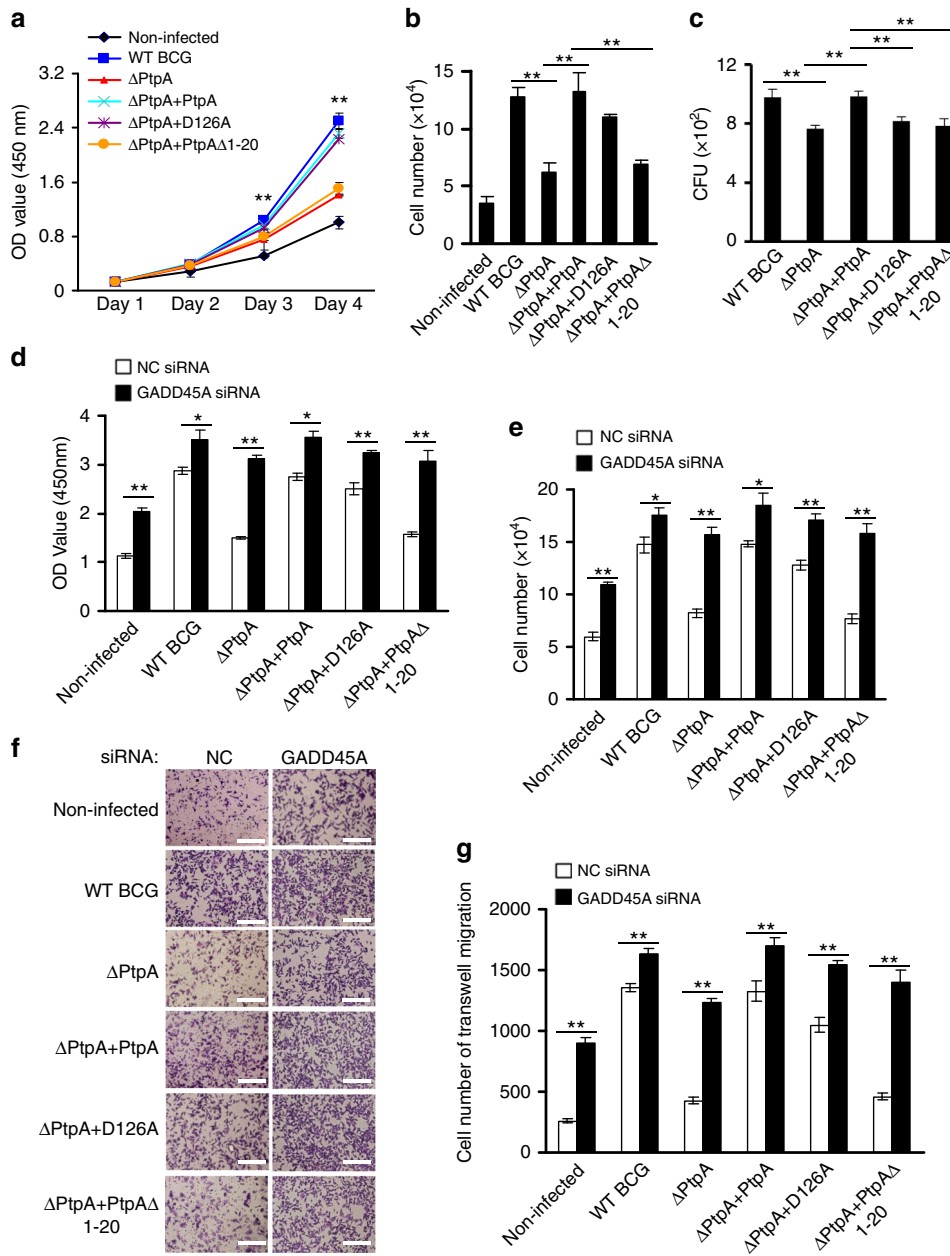

**Fig. 9** PtpA promotes A549 cell proliferation and migration in a DNA-binding-dependent manner. **a**, **b**. CCK-8 assay of A549 cells. Cells were infected with WT BCG, BCG ΔPtpA, BCG (ΔPtpA + PtpA), BCG (ΔPtpA + D126A), or BCG (ΔPtpA + PtpA Δ1-20) at a MOI of 10 for 0–4 days, and then the OD450 value was measured and cell numbers were calculated. **c** Survival of BCG strains in U937 cells infected as in **a**, **b** for 4 days. **d**, **e** CCK-8 assay of A549 cells with GADD45A knock-down. Cells were transfected with negative control RNA (NC RNA) or GADD45A-specific siRNA, followed by infection with WT BCG, BCG ΔPtpA, BCG (ΔPtpA + PtpA), BCG (ΔPtpA + D126A), or BCG (ΔPtpA + PtpA Δ1-20) at a MOI of 10 for 96 h, and then the $OD_{450}$ value was measured and cell numbers were calculated. **f**, **g** Transwell migration assay of A549 cells with GADD45A knock-down. Cells were transfected with siRNAs and infected with BCG strains as in **d**, then the cells were allowed to migrate on transwell inserts for 12 h. The cells that had migrated through the filter into the lower wells were quantitated by Crystal Violet assay **f** and were expressed as the total cell numbers of lower wells **g**. Scale bars, 200 μm. *$P < 0.05$ and **$P < 0.01$ (unpaired two-tailed Student's t-test). Data are representative of one experiment with at least three independent biological replicates (**a–e**, **g**; mean and s.e.m., $n = 3$)

of the mycobacterial intracellular survival even more significantly than that of phosphatase-inactive PtpA D126A strain (Fig. 6b–d and Supplementary Fig. 7a, b), suggesting that the DNA-binding region of PtpA may possess additional immune-regulatory function, independent of the phosphatase activity of PtpA.

Both TNF and IL-1β have been demonstrated to play important roles in host defense against mycobacterial infection[22–24]. In order to determine whether the immune suppression function of nuclear PtpA during mycobacterial infection is dependent on TNF or IL-1β, we used bone marrow derived macrophages (BMDMs) from WT mice and from mice deficient in TNFα or IL-1β to examine the inflammatory cytokine expression and BCG intracellular survival during infection. Consistent with previous studies[22–24], our data showed increased

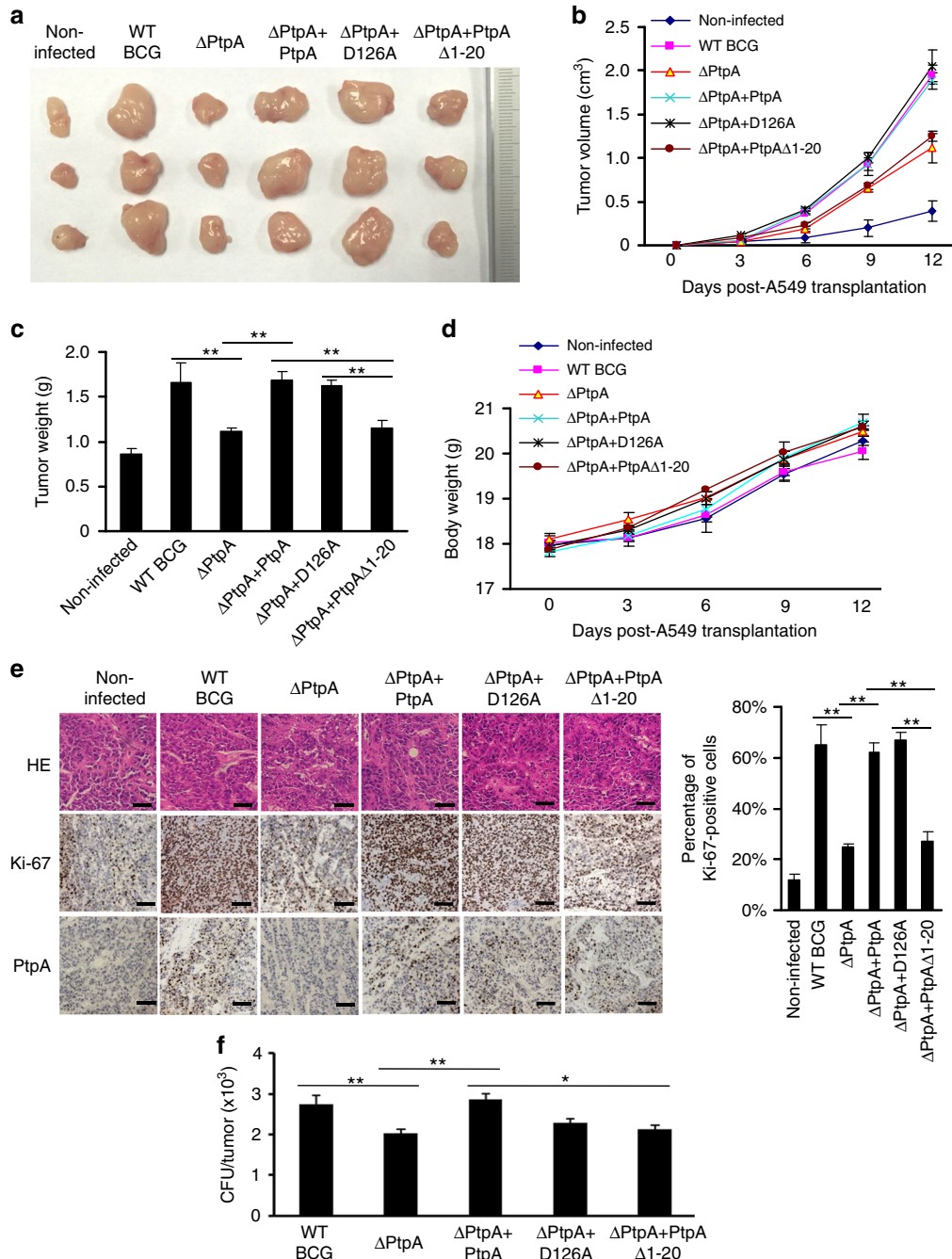

**Fig. 10** PtpA promotes tumor growth in nude mice. **a** Photograph of tumors. Tumors were derived from non-infected A549 cells or cells infected with WT BCG, BCG ΔPtpA, BCG (ΔPtpA + PtpA), BCG (ΔPtpA + D126A), or BCG (ΔPtpA + PtpA Δ1-20) in BALB/c nude mice. **b** Tumor growth curve in nude mice. After tumor cells were injected subcutaneously into the right armpit of nude mice, the short and long diameters of the tumors were measured every 3 days and tumor volumes (cm$^3$) were calculated. **c** Weight of tumors in nude mice. **d** Time course of the body weight of the mice. **e** Representative hematoxylin-eosin (*H&E*) staining histopathologic images of tumor tissues from mice (*upper panels*) and the immunohistochemical analysis of Ki-67 (*middle panels*) and PtpA in tumors (*lower panels*). *Scale bars*, 200 μm. *Right*, percentage of Ki-67-positive cells in the tumors. About 200 cells were counted. **f** Bacterial load in homogenates of tumors from nude mice killed 12 days after tumor inoculation. The average number of CFU per tumor was calculated. *$P < 0.05$ and **$P < 0.01$ (unpaired two-tailed Student's *t*-test). Data are representative of one experiment with two independent biological replicates (**b–f**; mean and s.e.m. of $n = 6$ mice per group)

intracellular bacterial loads in $TNF\alpha^{-/-}$ and $IL1\beta^{-/-}$ BMDMs compared with WT BMDMs. In addition, we found that compared with WT BMDMs, $TNF\alpha^{-/-}$ BMDMs exhibited higher levels of $Il1b$ mRNAs, while $IL1\beta^{-/-}$ BMDMs exhibited higher levels of $Tnf$ mRNAs, which might be caused by higher bacterial loads in $TNF\alpha^{-/-}$ and $IL1\beta^{-/-}$ BMDMs. Furthermore, we observed that the inhibition of $Il1b$ or $Tnf$ expression as well as the promotion of mycobacterial intracellular survival mediated by nuclear PtpA were partially abrogated in $TNF\alpha^{-/-}$ and $IL1\beta^{-/-}$ BMDMs, compared with WT BMDMs, during BCG infection (Fig. 7a–c and Supplementary Fig. 8a, b). Collectively, these results indicate that the nuclear PtpA-mediated immune suppression response is partially dependent on TNF and IL-1β.

To further confirm the immune suppression function associated with the DNA-binding ability of PtpA, we challenged C57BL/6 mice intratracheally with different BCG strains, i.e., WT BCG, BCG ΔPtpA, BCG (ΔPtpA + PtpA), BCG (ΔPtpA + D126A) and BCG (ΔPtpA + PtpA Δ1-20). Consistent with our previous study[9], deletion of $ptpA$ in BCG increased the levels of $Tnf$, $Il1b$, and $Il12b$ mRNAs in the spleens, reduced the bacterial loads in the lungs, and inhibited the cellular infiltration in the lungs and livers of the infected mice (Fig. 8a–f). Consistently, mice infected with BCG (ΔPtpA + PtpA Δ1-20) exhibited slightly more mRNA levels of $Il1b$ and $Il12b$ in the spleen and lower bacterial load in the lungs than those infected with BCG (ΔPtpA + D126A). Furthermore, both BCG (ΔPtpA + PtpA Δ1-20) and (ΔPtpA + D126A) strains reduced cellular infiltration in the lungs and livers in infected mice (Fig. 8a–f). Taken together, our data suggest that both phosphatase activity and DNA-binding ability of PtpA contribute to its role in the suppression of host innate immune responses to mycobacteria.

**PtpA promotes in vitro proliferation of infected cancer cells**. Since $GADD45A$, a direct target of PtpA, plays an important role in regulating cell proliferation and migration[25, 26], we thus performed CCK-8 and cell migration assays to examine the regulatory roles of PtpA on proliferation and migration of the human lung adenoma A549 cells infected with WT BCG, BCG ΔPtpA, BCG (ΔPtpA + PtpA), BCG (ΔPtpA + D126A), or BCG (ΔPtpA + PtpA Δ1-20). We found that PtpA promoted the proliferation and migration of BCG-infected A549 cells, likely by binding to target DNA in a phosphatase-independent manner, since the deletion of the DNA-binding region of PtpA, but not the phosphatase-inactivating mutation D126A, abolished this function (Fig. 9a–g). Knock-down of $GADD45A$ alone also promoted proliferation and migration of A549 cells. The combination of $GADD45A$ knock-down and BCG infection (with either WT or PtpA mutant strains) showed a slight synergy in promoting cell proliferation and migration (Fig. 9d–g). Thus, our results suggest that PtpA may promote cell proliferation and migration, partially through targeting $GADD45A$. However, our data do not exclude the possibility that additional mycobacterial effector proteins may also promote cell proliferation and migration. Also, PtpA might regulate host cell proliferation and migration through targeting other host genes.

**PtpA-expressing BCG promotes tumor growth in a mouse xenograft model**. To investigate the potential regulatory function of PtpA in tumor progression in vivo, we used a mouse xenograft model using A549 cells infected with WT BCG, BCG (ΔPtpA + PtpA), BCG ΔPtpA, BCG (ΔPtpA + D126A), or BCG (ΔPtpA + PtpA Δ1-20) strain. Non-infected A549 cells were used as control. Subcutaneous tumor formation was observed in all nude mice 6 days after injection. BCG ΔPtpA or BCG (ΔPtpA + PtpA Δ1-20) infection significantly decreased the size and weight of the

xenograft tumor as compared with that of WT BCG, BCG (ΔPtpA + PtpA) and BCG (ΔPtpA + D126A) groups at 9 and 12 days after injection (Fig. 10a–c). The weight of the mice exhibited no significant differences among the six groups (Fig. 10d). Furthermore, tumor tissues were embedded in paraffin and further analyzed with hematoxylin-eosin staining and immunohistochemistry staining to detect the levels of Ki-67 and PtpA (Fig. 10e), and the bacterial load in homogenates of tumors was calculated to examine the survival of BCG in the mice (Fig. 10f). The expression of Ki-67, a measure for tumor cell proliferation, was increased by PtpA or its phosphatase-inactive mutant D126A in xenograft tumors, in comparison with the truncated PtpA (PtpA Δ1-20). These results suggest that PtpA-expressing mycobacteria may promote growth of infected cancer cells in vivo, in a phosphatase-independent manner.

We summarized our findings in this study with a model to depict the role that PtpA plays in the nucleus of host cells during mycobacterial infection (Supplementary Fig. 9). Nuclear PtpA may regulate the transcription of certain important host genes involved in innate immunity as well as cell proliferation and migration.

**Discussion**

We previously reported that PtpA inhibits JNK/p38 MAPK and NF-κB signaling pathways and suppresses host innate immunity by co-opting the host ubiquitin system in the cytoplasm of macrophages[9]. In this study, we find that PtpA is present also in the host cell nucleus, and can inhibit the transcription of $TNFRSF8$ and $MAP4K2$. TNFRSF8 is a member of the TNF-receptor superfamily. TRAF2 and TRAF5 can interact with this receptor and mediate the signal transduction that leads to the activation of NF-κB pathway[16, 27]. MAP4K2 is a member of the serine/threonine protein kinase family. It is activated by TNF and has been shown to specifically activate JNK and p38 signaling pathways via reduced activation of the MAPK-kinase-kinases (MAP3Ks) mixed lineage kinases (MLKs)-2 and -3, while ERK or NF-κB pathway is not affected by MAP4K2[17, 28–32]. In addition, another PtpA target, $GADD45A$, also responds to various physiological and environmental stressors by mediating activation of the p38/JNK pathway via MTK1/MEKK4 kinase[25]. Together, our results provide new insights into PtpA-mediated inhibition of the NF-κB and JNK/p38 signaling pathways by regulating the transcription of certain genes in the nucleus of host cells[9]. Thus, PtpA functions both in the cytoplasm and nucleus of host cells to suppress host innate immune signaling pathways. Other nuclear-translocated proteins (nucleomodulins) have previously been identified in bacterial pathogens, such as OspF from *Shigella flexneri*, LntA from *Listeria monocytogenes*, NUE from *Chlamydia trachomatis*, RomA from *Legionella pneumophila*, and AnkA from *Anaplasma phagocytophilum*[33–37].

In addition to targeting immune signaling pathways, pathogens including Mtb can modulate other innate immunity-associated cellular functions such as cell apoptosis[38, 39]. Our study identified a few potential PtpA target genes (such as $TNFRSF21$) that are involved in the regulation of apoptosis[20]. Furthermore, inhibition of the pathogen clearance process of host macrophages is also an important way to promote the intracellular survival of pathogens. After ingestion of pathogens, the subsequent successful maturation of the phagosome in macrophages is the key for the clearance of the pathogens. Thus the phagosome maturation process is also frequently targeted by effector proteins secreted by intracellular pathogens such as Mtb for their intracellular survival[40]. During the maturation process, the composition of the phagosomes is modified by recycling plasma membrane molecules and by sequentially acquiring markers of the early endosomes, as well as

late endosomes and lysosomes[41, 42]. This step-wise maturation process of phagosome involves a variety of Rab GTPases, which play important roles in various intracellular membrane and vesicle trafficking steps[43–45]. In this study, we identified several RAB family members, such as RAB11A and RAB37[19, 46–48] as potential targets of PtpA. Thus, we speculate that PtpA might also regulate the phagosome maturation process and inhibit pathogen clearance, partially through regulating the transcription of RAB family protein-coding genes such as *RAB11A* and *RAB37*.

It has been previously reported that lung carcinogenesis can be induced by chronic TB infection in a mouse model[49], and that BCG can promote the survival of A549 and several other tumor cells from TNFα-induced apoptosis thereby promoting tumor-igenesis in xenograft studies[50]. Furthermore, Mtb-infected THP-1 cells can induce epithelial mesenchymal transition (EMT) in the lung adenocarcinoma epithelial cell line A549[51]. However, causal links between Mtb infection and lung cancer in humans have not been demonstrated, and there is conflicting evidence concerning a possible association between Mtb-caused pulmonary TB and subsequent risk of lung cancer[52, 53]. In this study, we found that PtpA can inhibit the transcription of *GADD45A*, a gene encoding a protein involved in cell division, cell death and senescence, and DNA-damage repair. *GADD45A* knock-out mice display increased ionizing radiation-induced carcinogenesis[18, 26, 54], suggesting that the cell cycle might be out of control if the transcription of *GADD45A* is inhibited. Our data showed that infection with PtpA-expressing BCG can promote proliferation and migration of A549 cells, partially through targeting *GADD45A*. We also noticed that several potential PtpA-targeted ncRNA genes (such as miR-488, CASC2, and miR-622) are involved in tumor progression through regulating cell apoptosis, proliferation, and migration[55–57]. Thus, we speculate that PtpA might potentially contribute to lung cancer development during chronic Mtb infection through regulating the transcription of certain checkpoint protein-coding genes (such as *GADD45A*) and ncRNAs.

It should be mentioned that there is a discrepancy between the number of target genes detected by our ChIP-seq analysis and the results of our EMSA assays. Indeed, these two experiments are clearly different and may give different results. For example, due to the crosslinking step, ChIP-seq analysis may detect proteins indirectly binding to DNAs; in this scenario, PtpA might be part of a multi-protein complex, binding to other transcription factors and regulating their activities towards target genes in a phosphatase-dependent or -independent manner. Additionally, some important co-factors might be present in vivo (ChIP-seq analysis) but missing in vitro (EMSA).

In summary, this study reveals that mycobacterial PtpA plays roles in the nucleus of host cells by regulating the transcription of certain host genes involved in innate immunity, cell proliferation, and migration, either by directly binding to the promoters of its target genes, or in an indirect manner. However, further research is required to test whether there is any causal link between Mtb infection and lung cancer development in humans.

## Methods

**Bacterial strains, mammalian cell lines and antibodies.** Bacterial strains are listed in Supplementary Table 1. *E. coli* DH5α and BL21 were grown in flasks using lysogeny broth medium. *M. smegmatis* and *M. bovis* BCG strains were grown in Middlebrook 7H9 broth (7H9) supplemented with 10% oleic acid–albumin–dextrose–catalase (OADC) and 0.05% Tween-80 (Sigma), or on Middlebrook 7H10 agar (BD) supplemented with 10% OADC. *M. smegmatis* strains included WT *M. smegmatis* (WT-*M. smegmatis*), PtpA-overexpressing *M. smegmatis* (PtpA-*M. smegmatis*), and PtpA D126A-overexpressing *M. smegmatis* (PtpA D126A-*M. smegmatis*). *M. bovis* BCG strains included WT BCG, PtpA-deleted BCG (BCG ΔPtpA), BCG ΔPtpA complemented with WT PtpA (ΔPtpA + PtpA), BCG ΔPtpA complemented with PtpA D126A (ΔPtpA + D126A), and BCG ΔPtpA complemented with PtpA Δ1-20 (ΔPtpA + PtpA Δ1-20)

(created as described previously[9]). Plasmids and oligonucleotides used in the study are listed in Supplementary Table 1. HEK293T (ATCC CRL-3216), U937 cells (ATCC CRL-1593.2), and A549 cells (ATCC CCL-185) were obtained from the American type culture collection (ATCC). HEK293T and A549 cells were cultured in Dulbecco's modified Eagle's medium (DMEM, Gibco) with 10% FBS and U937 cells were maintained in RPMI 1640 medium (Gibco) with 10% FBS. For infection, U937 cells were cultured for 24 h in culture medium supplemented with 10 ng/ml of Phorbol 12-myristate 13-acetate (PMA, Sigma). The cell lines were tested for mycoplasma contamination using MycoAlert Mycoplasma Detection Kit (Lonza, # LT07-418).

The following antibodies were used in this study: anti-Myc (sc-40, Santa Cruz), anti-tubulin (2148, Cell Signaling Technology), anti-PARP (9542, Cell Signaling Technology), anti-PtpA (prepared as described previously[9]), anti-IgG (2729, Cell Signaling Technology), and anti-Flag (14793, Cell Signaling Technology). pHrodo Red succinimidyl (NHS) ester used for bacteria staining was purchased from Invitrogen (P36600, Invitrogen).

**Mice.** *Tnf*[−/−] mice were from the Jackson Laboratories[58], *Il1β*[−/−] mice were described previously[59]. Both *Tnf*[−/−] and *Il1β*[−/−] mice were on C57BL/6 genetic background. WT C57BL/6 mice and BALB/c nu/nu mice were purchased from Vital River (Beijing, China). All mice were housed in a specific pathogen-free facility using standard humane animal husbandry protocols, which were approved by the animal care and use committee of the Institute of Microbiology (Chinese Academy of Sciences).

**Establishment of a U937 cell line stably expressing flag-PtpA.**
HEK293T cells were co-transfected with PMSCVpuro-Flag-PtpA, pVSV-G and pCMV-Gag-Pol vectors to obtain virus. At 48 h post-transfection, the medium was collected and filtered with a 0.45 µm filter. And then, the medium containing virus was used to infect U937 cells supplemented with polybrene. After 48 h, the medium was replaced with fresh RPMI medium containing 10% FBS and 10 µg/ml pur-omycin (s7417, Selleck). Two weeks later, the cells stably expressing Flag-PtpA were selected.

**Preparation and purification of antibodies against PtpB.** Rabbit antibodies against PtpB were prepared and purified as described previously[9]. A total of 5 mg GST-PtpB fusion protein was purified and solubilized in Freeund's complete adjuvant to inject rabbit. The antibody specific to PtpB was isolated by passaging the immunized rabbit serum on protein A agarose (Santa Cruz).

**Immunoblot analysis.** Cells were washed three times with $1 \times$ PBS and lysed on ice in lysis buffer (P0013, Beyotime). About 10–50 µg of total cell lysates were separated on 10% sodium dodecyl sulphate (SDS)-polyacrylamide gel electrophoresis gels. The gels were electroblotted onto PVDF (Millipore) and the blots were developed by Immobilon Western Chemiluminescent HRP Substrate (WBKLS0500, Millipore) and exposed to X-ray film (Kodak).

**Cell staining and confocal microscopy.** Cells were seeded on glass coverslips and transfected with Lipofectamine 2000 (Invitrogen) or infected with BCG strains. Afterward, cells were washed with PBS for three times and fixed with 4% paraformaldehyde for 10 min, permeabilized with 0.5% Triton X-100 in PBS for 10 min, blocked with 1% milk in PBS for 1 h, and labeled with DAPI. Confocal images were taken with a Leica SP8 confocal system.

**Cell fractionation.** U937 cells were differentiated with 10 ng/ml PMA for 24 h and infected with WT BCG, BCG ΔPtpA, BCG (ΔPtpA + PtpA), or BCG (ΔPtpA + D126A) strain for 4 h at a multiplicity of infection (MOI) of 10. Non-infected cells were prepared as control. Cells were washed and treated with the following steps separately. Firstly, cells were subjected to hypotonic lysis buffer containing 10 mM HEPES (pH 7.9), 1.5 mM MgCl$_2$, 10 mM KCl, 0.34 M sucrose, 10% glycerol, and 0.1% Triton X-100 supplemented with 1% protease inhibitors cocktail for 10 min at 4 °C, followed by centrifugation at 1300×*g* for 10 min. The supernatant was collected as cytosolic fraction. Pellet of uninfected U937 cells were analyzed as nucleus fraction. Pellet of infected U937 was resuspended in PBS buffer containing 0.1% SDS and subjected to centrifugation at 2500×*g* for 15 min at 4 °C. The supernatant was collected as nucleus fraction.

**ChIP-Seq analysis.** U937 cells stably expressing Flag-PtpA were seeded at $1 \times 10^7$ cells per 10 cm dish. Cells were crosslinked with 1% formaldehyde for 10 min. Glycine was then added for 5 min to a final concentration of 0.125 M, cells scrapped and collected. The pellet was resuspended in lysis buffer (5 mM Tris-HCl, pH 8.0, 85 mM KCl, 0.5% NP40 supplemented with protease inhibitors cocktail) and subjected to centrifugation to collect nuclear pellet. The pellet was resuspended in RIPA buffer (20 mM Tris-HCl, pH 7.5, 100 mM NaCl, 1 mM EDTA, 0.5% NP40, 0.5% Na-deoxycholate, 0.1% SDS supplemented with protease inhibitors cocktail). The nuclear extracts were sonicated. The chromatin was incubated overnight with anti-Flag antibody (Sigma) followed by four washes with LiCl wash buffer (100 mM Tris-HCl, pH 7.5, 500 mM LiCl, 1% NP40, 1% Na-deoxycholate) and 1 wash with

TE (10 mM Tris-HCl, pH 7.5, 1 mM EDTA, pH 8.0). DNA was eluted in Elution Buffer (1% SDS, 0.1 M NaHCO$_3$) for 1 h at 37 °C and used for sequencing library preparation. DNA sequencing was performed by RIBOBIO Co. Ltd (Guangzhou, China). Binding sites were identified using MACS1.4 with a chromatin input library as the control data set. MACS assigns every candidate peak an enrichment P value, and those below a user-defined threshold P value (default 10$^{-5}$) are reported as the final peaks. The ratio between the ChIP-seq tag count and $\lambda_{local}$ is reported as the fold-enrichment. ChIP-seq was performed by using two independent biological replicates.

**Quantitative real-time PCR.** U937 cells were infected with BCG strains at a MOI of 10. After 2 h, cells were washed with 1 × PBS for three times and cultured in fresh RPMI 1640 supplemented with 10 μg/ml gentamycin for additional hours. Total RNA was extracted from the infected U937 cells and was reverse-transcribed into cDNA using the Hieff First Strand cDNA Synthesis Super Mix (11103ES70, YEASEN). The cDNA was then analyzed by qRT-PCR using Hieff qPCR SYBR Green Master Mix (11203ES03, YEASEN) on ABI 7300 system (Applied Biosystems). Each experiment was performed in triplicates and repeated at least three times. Data were analyzed by the $2^{-\Delta\Delta CT}$ method and were normalized to the expression of the control gene Gapdh.

**Electrophoretic mobility shift assays.** Gene promoter regions were amplified from human genomic DNA by PCR and biotinylated using the Biotin 3′ End Labeling Kit (GS008, Beyotime) and annealed according to the manufacturer's instructions. For EMSA, 10 μg of PtpA protein was incubated with 10 nM biotin-labeled oligos in the binding buffer (GS009, Beyotime) for 20 min at room temperature. For competitive EMSA, increasing amounts of unlabeled positive or negative oligos were added into the reaction mixture at 2 h prior to the addition of the constant amount of the labeled positive oligos. In some reactions, anti-PtpA antibody was added at 2 h prior to the addition of labeled oligos. The reaction mixtures were resolved on 1 mm-thick 15% non-denaturing polyacrylamide gels and transferred to HybondTM-N + membranes (FFN13, Beyotime). The DNA oligomers were UV crosslinked to the membrane and the labeled probes were detected by the LightShift Chemiluminescent EMSA Kit (GS009, Beyotime).

**Transwell migration assays.** Cells ($3.0 \times 10^4$) were plated in the upper chamber of BD BioCoat Control Culture Inserts (24-well plates, 8-μm pore size). Serum-free culture medium was added into each upper chamber, and medium containing 10% FBS was added to each bottom chamber. Cells were incubated on the membranes for 12 h. After 12 h, the cells that migrated were fixed in ethanol (95%) and stained for 30 min in a 0.1% Crystal Violet solution in 1 × PBS buffer. Three independent visual fields were examined via microscopy to count the number of the cells that had moved to the bottom chamber.

**CCK-8 assays.** Cells were seeded at a density of 2000 cells per well in 96-well plates and incubated at 37 °C. An aliquot of 10 μl of CCK-8 (CK04, DOJINDO) was added into the wells and incubated for 2 h. The absorbance was measured at 450 nm to calculate the numbers of viable cells in each well. Each experiment was performed in triplicates and repeated at least three times.

**Luciferase reporter assays.** HEK293T cells were seeded at $2.5 \times 10^4$ cells per well in a 12-well plate and were co-transfected with 1 μg pGL3-basic-promoter vector or mock vector and vector encoding WT or deletion mutant PtpA (1 μg). We amplified −700 bp to + 100 bp from the TSS as promoter region for this assay. pRL-TK (100 ng) was used as an internal control. After 24 h, the cells were harvested and analyzed for firefly luciferase and renilla luciferase activity using the dual luciferase reporter assay kit (E1910, Promega).

**Infection of macrophages cells.** U937 cells were seeded and treated as described previously[9]. The cells were then infected with WT BCG, BCG ΔptpA, BCG (ΔptpA + PtpA), BCG (ΔptpA + D126A), or BCG (ΔptpA + PtpA Δ1-20) strain separately for 0–24 h. CFU counting and quantitative PCR and ELISA assay were performed as described previously[9].

**Mouse infection.** Male and female SPF C57BL/6 mice were 6–8-weeks old during the course of the experiments and were age- and sex-matched in each experiment. No additional randomization or blinding was used to allocate experimental groups. Sample size was based on empirical data from pilot experiments. Mice were allowed to acclimate for 1 week after arrival. The 36 mice were randomly divided into 6 groups ($n = 6$) and were intratracheally infected with BCGΔPtpA, WT BCG, BCG (ΔptpA + PtpA), BCG (ΔptpA + D126A), or BCG (ΔptpA + PtpA Δ1-20) strain separately, and data were analyzed as described previously[9]. All animal studies were approved by the Biomedical Research Ethics Committee of Institute of Microbiology (Chinese Academy of Sciences).

**Preparation of bone marrow derived macrophages.** BMDMs were obtained by flushing the tibia and femurs of mice. Cells were cultured in DMEM supplemented with 10% FBS, 100 U/ml penicillin, 100 μg/ml streptomycin and 30% L929 cell supernatant (containing macrophage colony-stimulating factor, M-CSF) for 4–6 days. Mature BMDMs were then plated in 12-well plates overnight, and then infected with BCG ΔPtpA, WT BCG, BCG (ΔptpA + PtpA), BCG (ΔptpA + D126A), or BCG (ΔptpA + PtpA Δ1-20) strain separately at a MOI of 10 for 0–48 h. Afterward, cells were harvested for CFU counting or quantitative PCR.

**Xenograft tumor model.** A549 cells were seeded at $6 \times 10^6$ cells per 10 cm plate for 12 h before infection. For infection, frozen BCG strains were thawed and centrifuged and the supernatant was removed. The pellet was resuspended in DMEM medium with 0.05% Tween-80 and vortexed for 10 s to disperse pellet. The A549 cells were infected with BCG strains at a MOI of 20. After 2 h, the medium was discarded, and cells were washed for three times with 1 × PBS and incubated again in the fresh DMEM medium supplemented with 10 μg/ml gentamicin for an addition of 12 h. Before injection, the A549 cells were washed three times with 1 × PBS and harvested. Male and female SPF 6–8-weeks old BALB/c nu/nu mice (Beijing Vital River Laboratory Animal Technology Co., Ltd. China) were allowed to acclimate for 1 week after arrival. The 36 mice were randomly divided in 6 groups ($n = 6$). All groups received subcutaneous injection of 200 ml 1 × PBS per mouse containing: (1) $6 \times 10^6$ non-infected A549 cells; (2) $6 \times 10^6$ A549 cells infected with BCG ΔPtpA; (3) $6 \times 10^6$ A549 cells infected with WT BCG; (4) $6 \times 10^6$ A549 cells infected with BCG (ΔptpA + PtpA); (5) $6 \times 10^6$ A549 cells infected with BCG (ΔptpA + D126A); (6) $6 \times 10^6$ A549 cells infected with BCG (ΔptpA + PtpA Δ1-20). Animals were monitored daily for changes in weight, tumor size and signs of any sickness. All the mice were sacrificed 12 days after tumor inoculation. Tumor growth was determined by caliper measurement of the length and width of the tumor mass. Tumor volumes were calculated by the modified ellipsoidal formula: $V = 1/2$ (length × width$^2$)[60].

**Histopathology and immunohistochemistry.** The tumors were fixed in 10% buffered-formalin solution and embedded in paraffin. For morphological analysis, hematoxylin-eosin (H&E) staining was performed on 3-mm-thick sections from paraffin-embedded tumor blocks. Immunohistochemistry were performed using proliferation marker rabbit monoclonal to Ki-67 (1:100, Abcam, Cambridge, UK) and purified PtpA antibodies (prepared as described previously[9]) in order to assess the tumor cells proliferation (the Ki-67 labeling index)[61] regulated by Mtb PtpA, followed by a goat anti-rabbit secondary antibody using 3,3′-diaminobenzidine tetrahydrochloride (DAB) reagents as substrate.

**Data analysis.** Peak finding and downstream data analysis were performed using HOMER[62], a software suite for ChIP-seq analysis, which was partially created to support this study. MEME motif analysis was used to identify PtpA consensus motifs[13]. DNA-binding residues of PtpA were predicted by DP-bind, a web server for sequence-based prediction of DNA-binding residues in DNA-binding proteins[21]. The protein–protein interaction network was established by the STRING (Search Tool for the Retrieval of Interacting Genes/Proteins) database version 10.0[14]. GO over-representative enrichment analysis was performed by PANTHER (Protein Analysis Through Evolutionary Relationships).

**Statistical analysis.** Statistical analysis between groups was performed by unpaired two-tailed Student's t-test. Data are presented as mean ± s.e.m. $P < 0.05$ or $P < 0.01$ was considered to be statistically significant.

**Data availability.** The ChIP-seq data have been deposited in the Gene Expression Omnibus (GEO) database under accession code GSE99712. The authors declare that all other relevant data supporting the findings of this study are available within the article and its Supplementary Information files, or from the corresponding author on request.

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

## Acknowledgements

This work was supported by research funding from the National Key Research and Development Program of China (Grant Nos. 2017YFD0500300 and 2017YFA0505900), the National Basic Research Programs of China (Grant No. 2014CB74440), the National Natural Science Foundation of China (Grant Nos. 81371769, 81571536 and 81571954), the Beijing Natural Science Foundation (Grant No. 5162021), the Strategic Priority Research Program of the Chinese Academy of Sciences (Grant No. XDPB03), and the Youth Innovation Promotion Association CAS (Grant No. Y12A027BB2).

## Author contributions

C.H.L. and J.W. conceived and designed the experiments. J.W., P.G., L.Q., D.Z., and Q.C. performed experiments. M.Z., R.Z., G.M., Y.I., and G.F.G. contributed new experimental materials and provided technical assistance in experiments. F.T. and J.W. performed the bioinformatics analysis; J.W. and C.H.L. analyzed the data and wrote the manuscript. All authors reviewed and approved the manuscript.

## Additional information

**Competing interests:** The authors declare no competing financial interests.

