## [Peer Review File · Nature Communications]

Reviewers' comments:

Reviewer #1 (Remarks to the Author):

In the present study the authors have aimed at investigating the nuclear role of mycobacterium tuberculosis (Mtb) PtpA, a secreted protein which dephosphorylates p-JNK, p-p38 d thereby leading to suppression of host innate immunity. Using genome wide ChIP-seq analysis a total of 280 genes were identified as potentially regulated by PtpA, with a large portion of these PtpA target genes involved in host innate immunity, cell proliferation and migration. Additionally, the authors demonstrate that both phosphatase activity and DNA-binding ability of PtpA are important in suppressing host innate immune responses. Furthermore, it is shown that PtpA, independent of its phosphatase activity, promotes lung adenoma cell proliferation and migration in vitro and tumor growth in vivo, partially via binding to the promoter of GADD45A gene,.

Over all the work is well done and novel providing evidence that support a potential association between Mtb infection and lung cancer development.

Nevertheless, several issues to be addressed would increase impact:

1. In Figure 9, the authors must quantify the Ki-67 staining.
2. In Figure 9, the authors must stain the tumors for the ectopically expressed proteins - Δ PtpA etc..
3. In Figure 6, the authors show that deletion of PtpA promoted the production of TNF and IL-1 β and decreased bacterial survival in U937 cells. It will be interesting if the authors could use macrophages derived from TNF or IL1B deficient mice and check if the immune suppression response of Nuclear Mtb PtpA to mycobacteria is dependent on TNF or IL1B.

Reviewer #2 (Remarks to the Author):

Wang et. al. suggest that an M. tuberculosis protein is migrating to the host cell nucleus. They further claim that the protein regulate 280 host genes and identified a specific one and shown that it is involved in promoting tumor cells proliferation.

Unfortunately, the paper suffers from many major deficiencies that do not rule out an experimental artifact. First they use confocal microscopy to show migration of PtpA to the host cell nucleus. They use antibodies against PtpA to show when bacteria express PtpA it is co localized to the nucleus. PtpA is expressed in very low levels upon infection. Studies that used electron microscopy to co localize PtpA and host protein did not detect this protein in the nucleus of the macrophage. The authors did not use any other BCG protein to rule out bacterial protein migration and their microscopy did not detect PtpA in the cytosol. The microscopy images are technically poor and do not show mycobacteria or phagosomes. There is no time dependency or association with PtpA expression. The bar graph in figure 1 shows 0% co-localization of PtpA with the nucleus when PtpA was not present in the assay?! Obviously, you don't have co-localization if one of the objects is missing. The same stands for the Western. Both assays (western and Microscopy) don't rule out artificial affinity. Why the mutant is complemented with D126 and not WT PtpA?

The global analysis of genes that are regulated by PtpA is overwhelming yet the degree of regulation is not clear. QPCR show that it is a relatively minor effect and a mutant and delta PtpA complemented with active PtpA is again missing. In this case it is a crucial control to show the effect of active PtpA. The fact the only one region (GADD45A) was bound to PtpA suggest that all the rest are false positives.

Lastly, the authors showed that BCG is involved in tumor cell proliferation through PtpA. They also quote case reports showing some association between mycobacteria and cancer progression. Yet, a large body of solid literature points to the contrary. BCG is used to treat bladder and prostate cancer and is approved by the FDA as a biological drug against these indications.

Reviewer #3 (Remarks to the Author):

In this work Wang and colleagues investigate the role of PtpA, a mycobacterial secreted protein, in host cell transcriptional control, innate immunity regulation and tumor development. They report the identification of 280 host genes controlled by PtpA, among which there are components of the signaling pathways, cell apoptosis, cell growth, proliferation and migration. They study the impact of the phosphatase activity of PtpA on the regulation of gene expression and carry out *ex vivo* and *in vivo* experiments to define the biological role of PtpA.

The manuscript is potentially interesting, especially for the mycobacterial research community. However, there are several points that require attention:

1. Introduction page 3 «The treatment of active TB....which often leads to a poor compliance due to the emergence of multidrug-resistant strains of Mtb». As it is written now, it seems that the poor compliance is due to the emergence of MDR strains, while it is the opposite. This sentence needs rewording.

2. Introduction page 4. There is a long summary of the manuscript in the Introduction section (lines 75-94). This is not necessary and should be shortened. More information on PtpA could be added. For instance, has the structure been solved? Are there any important residues? Any essential domains? An Mtb mutant in *ptpA* exists: what is the phenotype?

3. I noticed several typos and grammar mistakes in the text, which should be thoroughly revised. For example «stain» instead of «strain».

4. Results page 6. The paragraph entitled «Genome-wide ChIP-seq analysis of Mtb PtpA-binding sites» should be re-written as it is unclear how many binding sites for PtpA were detected by ChIP-seq, how many of these were localized in intra- or inter-genic regions, how many were found to be associated with protein coding genes etc. In addition, I do not understand what the Authors mean with «3351 Mtb PtpA-specific transcripts were enriched». ChIP-seq does not identify transcripts but binding sites. Again «half of the transcripts were located in the promoter regions»: what does this sentence mean? Did the Authors carry out ChIP-seq or RNA-seq analysis? Which threshold did the Authors choose for identifying the targets in ChIP-seq analysis? The enrichment factor for each peak should be reported in the Supplementary Tables. I find the percentages misleading as I cannot understand how the Authors reduced the number of signals (3351) to 280 in the next paragraph.

5. Legend to Figure 2b: «Biotype distribution of all the transcripts....». This is unclear: ChIP-seq identifies binding sites and not transcripts.

6. Results page 6 «...the Input DNA (negative control)». In ChIP-seq experiments, the Input DNA is not the negative control. The Input DNA represents the sequencing depth throughout the DNA and is a normalizing factor rather than a negative control.

7. Results page 6 «We then analyzed the 280 proteins whose promoter regions...». Proteins do not have promoter regions. Genes have promoter regions and are transcribed into RNA, which is then translated into proteins. This sentence should be corrected. Also, how did the Authors select these 280 targets? Are they the most highly enriched in ChIP-seq experiments? If so, the enrichment factor should be provided in the Supplementary Tables.

8. Results page 8 «These results indicate that PtpA is located at the promoter regions of those genes...». This is a very strong statement that is contradicted on the following page, when results of EMSA experiments are presented. The impact on transcriptional activity is not sufficient *per se* to conclude that PtpA binds directly to the promoter region. A transcription factor may have an indirect effect, for instance through a regulatory cascade or through binding to a protein complex. Indeed, on the next page (page 9) The EMSA assay demonstrates binding to GADD45A only.

9. Discussion page 14 «We thus in this study.....from a new perspective». This long sentence should be reworded as it is unclear in the present format.
10. Discussion page 14 «...transcription of certain proteins..». Proteins are not transcribed. Genes are transcribed into RNA.
11. There is no discussion of the discrepancy between the number of targets detected by ChIP-seq and the results of the EMSA assay. The two experiments are clearly different and may give different results. However, there might be missing co-factors in vitro (EMSA), which are present in vivo (ChIP). Additionally, PtpA may be part of a multiprotein complex which controls transcription. In this case, PtpA may dephosphorylate other transcription factors by direct binding to these. The same contradiction is evident between the data reported on page 7 («We found that, except for GAS1, all the other seven promoters of the selected genes were targeted by PtpA») and those on page 9 («...only the GADD45A promoter region could be bound by Mtb PtpA directly»). How do the Authors explain these discrepancies?
12. Is it possible to identify the PtpA consensus sequence?
13. The Authors mention that some PtpA targets were found to be associated with non-coding RNA (ncRNA). Is any of these ncRNA involved in innate immunity/apoptosis/cell cycle control?
14. Methods. How many times was the ChIP-seq experiment performed? How was the enrichment calculated? Which threshold was applied?

Point-by-point response to the referees' comments

Reviewers comments:

Reviewer #1 (Remarks to the Author):

In the present study the authors have aimed at investigating the nuclear role of *Mycobacterium tuberculosis* (Mtb) PtpA, a secreted protein which dephosphorylates p-JNK, p-p38 d thereby leading to suppression of host innate immunity. Using genome wide ChIP-seq analysis, a total of 280 genes were identified as potentially regulated by PtpA, with a large portion of these PtpA target genes involved in host innate immunity, cell proliferation and migration. Additionally, the authors demonstrate that both phosphatase activity and DNA-binding ability of PtpA are important in suppressing host innate immune responses. Furthermore, it is shown that PtpA, independent of its phosphatase activity, promotes lung adenoma cell proliferation and migration in vitro and tumor growth in vivo, partially via binding to the promoter of GADD45A gene. Over all the work is well done and novel providing evidence that support a potential association between Mtb infection and lung cancer development.

R: We thank the reviewer for the encouraging comments on our manuscript.

Nevertheless, several issues to be addressed would increase impact:

1. In Figure 9, the authors must quantify the Ki-67 staining.

R: We thank the reviewer for reminding us this issue. We quantified the Ki-67 staining by calculating the percentage of Ki-67-positive cells in tumors as shown in revised Fig. 10e.

2. In Figure 9, the authors must stain the tumors for the ectopically expressed proteins - Δ PtpA etc.

R: We thank the reviewer for this suggestion. We repeated the immunohistochemical analysis using the PtpA antibody to detect the ectopically expressed PtpA in tumors (revised Fig. 10e). It should be mentioned that not like macrophages, epithelial cells (such as A549 cells) do not provide the optimum conditions for pathogen survival during mycobacterial infection. Thus, in this experiment, we observed gradually decreased intracellular survival of mycobacteria in A549 cells-derived tumors overtime. By the time of the immunohistochemical analysis, the tumors had grown for about two weeks in mice, thus the bacterial load was relatively low at that point, leading to the relatively low staining of the secreted protein PtpA at that time.

3. In Figure 6, the authors show that deletion of PtpA promoted the production of TNF and IL-1 β and decreased bacterial survival in U937 cells. It will be interesting if the authors could use macrophages derived from TNF or IL1B deficient mice and check if the immune suppression response of Nuclear Mtb PtpA to mycobacteria is dependent on TNF or IL-1 β .

R: We thank the reviewer for this suggestion. We analyzed mRNA levels of *Tnf* and *Il1b* as well as bacterial survival in mycobacteria-infected bone marrow-derived macrophages (BMDMs) from WT, *Tnf*^{-/-} and *Il1 β* ^{-/-} mice, and we found that nuclear PtpA-mediated immune suppression response is partially dependent on TNF and IL-1 β (revised Fig. 7 and Supplementary Fig. 8).

Reviewer #2 (Remarks to the Author):

Wang et. al. suggest that an *M. tuberculosis* protein is migrating to the host cell nucleus. They further claim that the protein regulates 280 host genes and identified a specific one and shown that it is involved in promoting tumor cells proliferation. Unfortunately, the paper suffers from many major deficiencies that do not rule out an experimental artifact. First they use confocal microscopy to show migration of PtpA to the host cell nucleus. They use antibodies against PtpA to show when bacteria express PtpA it is colocalized to the nucleus. PtpA is expressed in very low levels upon infection. Studies that used electron microscopy to colocalize PtpA and host protein did not detect this protein in the nucleus of the macrophage. The authors did not use any other BCG protein to rule out bacterial protein migration and their microscopy did not detect PtpA in the cytosol. The microscopy images are technically poor and do not show mycobacteria or phagosomes. There is no time dependency or association with PtpA expression. The bar graph in figure 1 shows 0% co-localization of PtpA with the nucleus when PtpA was not present in the assay?! Obviously, you don't have co-localization if one of the objects is missing. The same stands for the Western. Both assays (western and Microscopy) don't rule out artificial affinity. Why the mutant is complemented with D126 and not WT PtpA?

R: We thank the reviewer's encouraging comments and valuable suggestions. We agree with the reviewer that PtpA is expressed in very low levels upon infection. While repeating the confocal microscopy experiments, we tried to optimize the experimental conditions to improve the data quality. We also stained bacteria (red) and used another Mtb secreted protein tyrosine phosphatase B (PtpB) as a control in the analysis. Our data indicated that PtpA, but not PtpB, was largely co-localized with the nuclei of mycobacteria-infected macrophage cells (revised Fig. 1a and Fig. 1d). We also did a time course analysis for intracellular localization of PtpA and found that the nuclear localization of PtpA increased

over time within 6 h post-infection (revised Fig. 1b).

We also agree with the reviewer that the Δ PtpA strain complemented with WT PtpA should be included as a control, and we thus repeated all relevant experiments in which the required control was included (revised Fig. 1, Fig. 4c, Figs. 6-10, and Supplementary Figs. 7, 8)

The global analysis of genes that are regulated by PtpA is overwhelming yet the degree of regulation is not clear. QPCR show that it is a relatively minor effect and a mutant and delta PtpA complemented with active PtpA is again missing. In this case it is a crucial control to show the effect of active PtpA. The fact the only one region (*GADD45A*) was bound to PtpA suggest that all the rest are false positives.

R: We agree with the reviewer that the Δ PtpA strain complemented with active PtpA is a crucial control to show the effect of active PtpA, and we included this control while repeating the Quantitative PCR analysis as well as other relevant experiments (revised Fig. 1, Fig. 4c, Figs. 6-10, and Supplementary Figs. 7, 8).

We identified a total of 280 protein-coding genes potentially regulated by Mtb PtpA in the ChIP-seq analysis. PtpA may directly bind to the promoter region of its target genes to regulate their expression, or it may regulate target genes through an indirect way. For example, through a regulatory cascade or binding to a transcription factor-containing protein complex. In this paper, we verified that the promoter of *GADD45A* was directly bound by PtpA, while the other five genes (*MAP4K2*, *RNF187*, *TNFRSF8*, *TLR7* and *SLC35B2*) may be regulated by PtpA through an indirect manner. We have provided the discussion on the discrepancy between the number of targets detected by ChIP-seq and the results of the EMSA assay in the 4th paragraph of the Discussion part of the revised manuscript.

Lastly, the authors showed that BCG is involved in tumor cell proliferation through PtpA. They also quote case reports showing some association between mycobacteria and cancer progression. Yet, large body of solid literature points to the contrary. BCG is used to treat bladder and prostate cancer and is approved by the FDA as a biological drug against these indications.

R: We thank the reviewer for raising this question. We have repeated our experiments for many times and have obtained consistent results to demonstrate that BCG is involved in tumor cell proliferation in our experimental system.

Cancer arises and is fueled through a multi-step process involving deregulation of multiple signaling pathways, which control cellular growth and proliferation. Infections caused by a variety of bacterial pathogens may simply

contribute one such step, since many cellular signaling pathways and functions are targeted by certain bacterial pathogens (Vogtmann et al., *British Journal of Cancer*, 2016; Scanu et al., *Cell Host & Microbe*, 2015; Brenner et al, *PloS ONE*, 2011).

We noticed that, on the one hand, it has long been speculated that pulmonary tuberculosis is closely linked to lung cancer (Sakuraba M et al., *Ann Thorac Cardiovasc Surg*, 2006; Brenner et al, *PloS ONE*, 2011; Tian Y et al., *Biomed Res Int* 2015), and on the other hand, BCG has been used to treat bladder and prostate cancers. But the optimal methodology of BCG treatment for bladder cancers has been controversial, and one of the main limitations for such treatment is the local and systemic side effects ranging from cystitis, epididymitis, prostatitis to lung infection, liver toxicity and sepsis, and those side effects are dose dependent (Astram A et al., *Acta Med Indones.*, 2014). Furthermore, the effects of BCG treatment may vary in a context-dependent manner, and increasing epidemiological studies have indicated that this treatment could lead to poor treatment outcomes under certain circumstances. For example, a study reported a high complication rate of 81.5% for maintenance intravesical instillation therapy with BCG for non-muscle invasive bladder cancer (Ikeda M et al., *Hinyokika Kyo*. 2013). Another study made a conclusion that maintenance therapy with 3-monthly BCG for 3 years is not superior to standard induction therapy in high-risk non-muscle-invasive urothelial bladder carcinoma (Martinez-Pineiro L et al., *Eur Urol*. 2015). In addition, there are also studies demonstrating that the tumor microenvironment seems to influence the therapeutic response to BCG. For instance, a study found that increased infiltration of tumor associated macrophages is associated with poor prognosis of bladder carcinoma in situ after intravesical BCG instillation (Takayama H et a., *J Urol.*, 2009). Another study reported that tumor-infiltrating immune cell subpopulations influence the oncologic outcome after intravesical BCG therapy in bladder cancers (Pichler R et al., *Oncotarget*, 2016). Thus, dependent on host immune status, tumor type and stage, as well as the treatment methodology and dosing regimen, the BCG treatment might inhibit or promote the tumor progression. The detailed molecular mechanisms underlying the complex relationships among infection, host response and tumor growth are not fully understood and warrant more in-depth investigations.

Reviewer #3 (Remarks to the Author):

In this work Wang and colleagues investigate the role of PtpA, a mycobacterial secreted protein, in host cell transcriptional control, innate immunity regulation and

tumor development. They report the identification of 280 host genes controlled by PtpA, among which there are components of the signaling pathways, cell apoptosis, cell growth, proliferation and migration. They study the impact of the phosphatase activity of PtpA on the regulation of gene expression and carry out ex vivo and in vivo experiments to define the biological role of PtpA.

R: We thank the reviewer for the encouraging and insightful comments on our manuscript.

The manuscript is potentially interesting, especially for the mycobacterial research community. However, there are several points that require attention:

1. Introduction page 3 «The treatment of active TB....which often leads to a poor compliance due to the emergence of multidrug-resistant strains of Mtb». As it is written now, it seems that the poor compliance is due to the emergence of MDR strains, while it is the opposite. This sentence needs rewording.

R: We thank the reviewer for pointing this out. We rephrased this sentence in the revised manuscript.

2. Introduction page 4. There is a long summary of the manuscript in the Introduction section (lines 75-94). This is not necessary and should be shortened. More information on PtpA could be added. For instance, has the structure been solved? Are there any important residues? Any essential domains? An Mtb mutant in PtpA exists: what is the phenotype?

R: We thank the reviewer for this good suggestion. We shortened the summary of the manuscript and provided more information on PtpA (such as the crystal structure, important residues, essential domains, as well as the mutant phenotype of PtpA) in the revised manuscript.

3. I noticed several typos and grammar mistakes in the text, which should be thoroughly revised. For example «stain» instead of «strain».

R: We thank the reviewer for reminding us of this issue. We have proofread our manuscript to correct the typos.

4. Results page 6. The paragraph entitled «Genome-wide ChIP-seq analysis of Mtb PtpA-binding sites» should be re-written as it is unclear how many binding sites for PtpA were detected by ChIP-seq, how many of these were localized in intra- or inter-genic regions, how many were found to be associated with protein coding genes etc. In addition, I do not understand what the Authors mean with «3351 Mtb PtpA-specific transcripts were enriched». ChIP-seq does not identify transcripts but binding sites. Again «half of the transcripts were located in the promoter regions»:

what does this sentence mean? Did the Authors carry out ChIP-seq or RNA-seq analysis? Which threshold did the Authors choose for identifying the targets in ChIP-seq analysis? The enrichment factor for each peak should be reported in the Supplementary Tables. I find the percentages misleading as I cannot understand how the Authors reduced the number of signals (3351) to 280 in the next paragraph.

R: We thank the reviewer for raising these questions. We rewrote this paragraph and rephrased those confusing sentences in the revised manuscript accordingly. In this study, a total of 3351 potential Mtb PtpA-binding sites were detected by ChIP-seq (Supplementary Data 1), of which 46.79% (1568) were localized in intergenic regions, 35.21% (1180) were localized in intragenic regions (Supplementary Data 1 and Fig. 2a). Among the 3351 potential PtpA-binding sites analyzed, about 73.23% (2454) were identified as protein-coding associated regions, while the remaining ones were classified as different types of noncoding RNAs (ncRNAs) (20.68%, 693) and pseudogenes (5.34%, 179) (Supplementary Data 1 and Fig. 2b).

As pointed out by the reviewer that ChIP-seq does not identify transcripts but binding sites, we are sorry to cause the confusion by saying “3351 Mtb PtpA-specific transcripts were enriched”, and we corrected the typo in the revised manuscript as follows: “3351 potential Mtb PtpA-binding sites were detected”.

In this study, we identified ChIP-seq enrichment using Model-based Analysis of ChIP-seq (MACS). MACS assigns every candidate peak an enrichment P value, and those below a user-defined threshold P value (default 10^{-5}) are reported as the final peaks. The ratio between the ChIP-seq tag count and λ_{local} is reported as the fold-enrichment (which has been provided for each peak in the revised Supplementary Data 1 and 2).

As to the question regarding “How the authors reduced the number of signals (3351) to 280”, our answer is as follows: Among the 2454 protein-coding associated regions potentially bound by Mtb PtpA, 280 (11.41%) occurred in the promoter region ($\pm 2\text{kb}$ from the TSS) of known RefSeq genes, suggesting a transcription regulatory function of PtpA towards those genes (3rd paragraph in the Results part of the revised manuscript; Supplementary Data 2).

5. Legend to Figure 2b: «Biotype distribution of all the transcripts....». This is unclear: ChIP-seq identifies binding sites and not transcripts.

R: We thank the reviewer for reminding us of this issue. We rephrased those sentences in the revised manuscript.

6. Results page 6 «...the Input DNA (negative control)». In ChIP-seq experiments, the Input DNA is not the negative control. The Input DNA represents the sequencing depth throughout the DNA and is a normalizing factor rather than a negative control.

R: We thank the reviewer for correcting this faulty usage. Indeed, the Input DNA is a normalizing factor rather than a negative control in ChIP-seq experiments. We have corrected this mistake accordingly.

7. Results page 6 «We then analyzed the 280 proteins whose promoter regions...». Proteins do not have promoter regions. Genes have promoter regions and are transcribed into RNA, which is then translated into proteins. This sentence should be corrected. Also, how did the Authors select these 280 targets? Are they the most highly enriched in ChIP-seq experiments? If so, the enrichment factor should be provided in the Supplementary Tables.

R: We thank the reviewer for pointing out that genes, but not proteins, have promoter regions. We have corrected this mistake in the revised manuscript.

As is described above, among the 2454 protein-coding associated regions potentially bound by Mtb PtpA, 280 (11.41%) occurred in the promoter region (\pm 2kb from the TSS) of known RefSeq genes, suggesting a transcription regulatory function of PtpA towards those genes (3rd paragraph in the Results part of the revised manuscript; Supplementary Data 2).

The fold-enrichment value for each binding site from ChIP-seq data were provided in the revised Supplementary Data 1 and 2.

8. Results page 8 «These results indicate that PtpA is located at the promoter regions of those genes...». This is a very strong statement that is contradicted on the following page, when results of EMSA experiments are presented. The impact on transcriptional activity is not sufficient per se to conclude that PtpA binds directly to the promoter region. A transcription factor may have an indirect effect, for instance through a regulatory cascade or through binding to a protein complex. Indeed, on the next page (page 9) The EMSA assay demonstrates binding to GADD45A only.

R: We thank the reviewer for pointing out this issue. Indeed, the effect on transcriptional activity is not sufficient per se to conclude that PtpA binds directly to the promoter regions of its target genes, and a transcription factor may have an indirect effect. We have rephrased those inappropriate sentences.

9. Discussion page 14 «We thus in this study.....from a new perspective». This long sentence should be reworded as it is unclear in the present format.

R: We thank the reviewer for pointing this out, and we have reworded this sentence.

10. Discussion page 14 «...transcription of certain proteins.». Proteins are not transcribed. Genes are transcribed into RNA.

R: We thank the reviewer for reminding us of this issue, and we have corrected this mistake.

11. There is no discussion of the discrepancy between the number of targets detected by ChIP-seq and the results of the EMSA assay. The two experiments are clearly different and may give different results. However, there might be missing co-factors in vitro (EMSA), which are present in vivo (ChIP). Additionally, PtpA may be part of a multiprotein complex which controls transcription. In this case, PtpA may dephosphorylate other transcription factors by direct binding to these. The same contradiction is evident between the data reported on page 7 («We found that, except for GAS1, all the other seven promoters of the selected genes were targeted by PtpA») and those on page 9 («...only the GADD45A promoter region could be bound by Mtb PtpA directly»). How do the authors explain these discrepancies?

R: We appreciate the reviewer's guiding opinions and constructive comments. We have provided the discussion of the discrepancy between the number of targets detected by ChIP-seq and the results of the EMSA assay in the revised manuscript (the last two paragraphs in the Discussion part of the revised manuscript).

In this study, we examined the regulatory function of PtpA towards 7 potential target genes involved in host innate immunity and cell proliferation (including *GAS1*, *MAP4K2*, *RNF187*, *GADD45A*, *TNFRSF8*, *TLR7* and *SLC35B2*), and we found that 6 of those genes with higher fold-enrichment (including *MAP4K2*, *RNF187*, *GADD45A*, *TNFRSF8*, *TLR7* and *SLC35B2*; fold-enrichment > 10) were indeed regulated by PtpA and 1 gene with lower fold-enrichment (*GAS1*; fold-enrichment = 5.11) was not regulated by PtpA. In addition, we verified that the promoter of *GADD45A* was directly bound by PtpA, while the other five genes (*MAP4K2*, *RNF187*, *TNFRSF8*, *TLR7* and *SLC35B2*) may be regulated by PtpA through an indirect manner.

12. Is it possible to identify the PtpA consensus sequence?

R: We thank the reviewer for raising this question. As shown in the revised Fig. 2e, two Mtb PtpA consensus motifs were identified by MEME motif analysis of the DNA sequences enriched in potential Mtb PtpA-binding regions (Fig. 2e).

The percentages of the motif-containing regions in potential PtpA targets are 46.86% and 14.7%, respectively.

13. The Authors mention that some PtpA targets were found to be associated with non-coding RNA (ncRNA). Is any of these ncRNA involved in innate immunity/apoptosis/cell cycle control?

R: We thank the reviewer for raising this question. We noticed that several potential PtpA-targeting ncRNAs (such as miR-488, CASC2 and miR-622, etc.) are involved tumor progression through regulating cell apoptosis, proliferation and migration. We discussed this in the revised manuscript (3rd paragraph in the Discussion part).

14. Methods. How many times was the ChIP-seq experiment performed? How was the enrichment calculated? Which threshold was applied?

R: We thank the reviewer for these questions. In this study, we performed two replicates of Mtb PtpA ChIP-seq experiments and identified their overlap results for further analysis. ChIP-seq data were presented as the mean of two biological replicates (revised Supplementary Data 1 and 2).

We identified ChIP-seq enrichment using MACS, which assigns every candidate peak an enrichment P value, and those below a user-defined threshold P value (default 10^{-5}) are reported as the final peaks. The ratio between the ChIP-seq tag count and λ_{local} is reported as fold-enrichment. We provided fold-enrichment value for each binding site from ChIP-seq data in the revised manuscript (revised Supplementary Data 1 and 2).

Once again, we greatly appreciate the reviewers for having helped us improve this manuscript tremendously.

REVIEWERS' COMMENTS:

Reviewer #1 (Remarks to the Author):

The issues raised have been addressed appropriately and the revised manuscript has been much improved.

Reviewer #2 (Remarks to the Author):

This paper describes a potential nuclear activity attributed to the mycobacterial secreted protein. Through a series of confocal microscopy and western analysis experiment the authors claim that PtpA is migrating to macrophages nucleus. I doubt these findings. The controls (figure S-1) show high background activity for the background and the use of confocal microscopy is not an ideal tool to detect migration to the nucleus. The western analysis image does not seem to come from the same gel for ptpA and the controls and as such are problematic. One cannot rule out high background or nonspecific presence of ptpA detected by antibody.

The non specific binding to too many genes is even more alarming and the significance of such activity and the potential havoc it may cause on the macrophage activity is not in line with the minor effect and current knowledge about the effect of Mtb or BCG on macrophage biology. Such a broad effect with limited significant effect should be discussed.

More alarming is the finding that BCG can cause cancer through PtpA. BCG vaccination is given to millions every year since the 1920's. It is also used by direct inoculation to treat bladder cancer. Over 90 years of experience did not show any correlation or risk associated with development of cancer. Any study that may cast doubt or risk on a well-established vaccination and/ or treatment should be careful, well designed and backed by evidence. This remark has been raised before and the authors fail to provide satisfactory answers. They do provide citations about the efficacy of BCG therapy and a single sporadic case reports associating lung cancer with TB (and not BCG) but do not provide sufficient evidence that provide link between BCG and Cancer. Experimentally they used a cancer model building on a cell line that is not a recognized model for mycobacterial infection and in a multiple of infectivity of 20! which may cause cell lysis and release of carcinogenic matter. Macrophages infected by BCG or Mtb remain viable and contain the infection. No information was provided as to the fate of BCG with these cell lines or the nude mice. The experiment should be validated before jumping into controversial declarations.

Reviewer #3 (Remarks to the Author):

The Authors satisfactorily addressed my concerns.

Point-by-point response to the referees' comments

Reviewers comments:

Reviewer #2 (Remarks to the Author):

This paper describes a potential nuclear activity attributed to the mycobacterial secreted protein. Through a series of confocal microscopy and western analysis experiment the authors claim that PtpA is migrating to macrophages nucleus. I doubt these findings. The controls (figure S-1) show high background activity for the background and the use of confocal microscopy is not an ideal tool to detect migration to the nucleus. The western analysis image does not seem to come from the same gel for ptpA and the controls and as such are problematic. One cannot rule out high background or nonspecific presence of ptpA detected by antibody. The nonspecific binding to too many genes is even more alarming and the significance of such activity and the potential havoc it may cause on the macrophage activity is not in line with the minor effect and current knowledge about the effect of Mtb or BCG on macrophage biology. Such a broad effect with limited significant effect should be discussed.

R: We thank the reviewer for raising these concerns. We performed confocal microscopy experiments with macrophages infected with mycobacterial strains (in Figure 1a, 1b) or transfected with the vector to overexpress PtpA (in previous Figure S1a, 1b). Data from all those experiments demonstrate that PtpA is present both in the cytoplasm and the nucleus of host cells. Since the GFP-tagged proteins tend to exhibit relatively high background activity in confocal microscopy experiments (as shown in previous Figure S1a, 1b, which have been changed to Figure S1b, 1c in the revised manuscript), we thus further added experiments using Myc-tagged PtpA (which exhibited lower background activity) to confirm the subcellular location of overexpressed PtpA (revised Figure S1a). In addition, we have carefully repeated the western analysis experiments and obtained consistent results. Taken together, our data confirmed that PtpA does indeed enter into the nucleus of host cells.

The reviewer indicated that the effect of Mtb/BCG on macrophage biology is minor. Actually, increasing studies have demonstrated that Mtb/BCG have developed multiple strategies to interact with macrophages and exert broad effects on macrophage biology. The following are just a few examples demonstrating that a variety of Mtb/BCG components can regulate multiple signaling pathways and cellular functions of the macrophages, thus promoting the survival and dissemination of Mtb/BCG in macrophages: 1) SapM, Ndk, ESAT-6 and PtpA arrest phagosome by targeting PI3P, Rab5/7 and V-ATPase, respectively (Sun et al., 2010; Wong et al., 2011; Puri et al., 2013; Chandra et al., 2015); 2) PE_PGRS47 reduces LC3 puncta and p62 degradation, whereas EsxH and EsxG impair MVB formation to inhibit autophagy (Saini et al., 2016); 3) PtpA competitively interacts with TAB3 to prevent the binding of K63 Ub chains

and directly dephosphorylates phos-p38 and phos-JNK to inhibit both NF- κ B and MAPK pathway (Wang et al., 2015); 4) MPT64, Rv3354 and PtpA have different apoptotic regulatory functions when delivered into cytoplasm (Liu et al., 2015); 5) ZMP1 acts as a Zn²⁺ metalloprotease with the ability to restrain the host inflammasome development and suppress the activity of caspase-1 (Master et al., 2008). In summary, we think that the broad effects potentially caused by the mycobacterial phosphatase PtpA (as indicated in this study and by others) fit well with the unfolding picture depicting the broad and pleiotropic effects of Mtb/BCG on macrophage biology.

More alarming is the finding that BCG can cause cancer through PtpA. BCG vaccination is given to millions every year since the 1920's. It is also used by direct inoculation to treat bladder cancer. Over 90 years of experience did not show any correlation or risk associated with development of cancer. Any study that may cast doubt or risk on a well-established vaccination and/ or treatment should be careful, well designed and backed by evidence. This remark has been raised before and the authors fail to provide satisfactory answers. They do provide citations about the efficacy of BCG therapy and a single sporadic case reports associating lung cancer with TB (and not BCG) but do not provide sufficient evidence that provide link between BCG and Cancer.

R: We thank the reviewer for raising this question again. The reviewer feels that it is difficult to accept the concept that BCG, which has been used as a vaccine since the 1920's, is associated with cancer. Actually, we do not claim that BCG alone can cause normal cells to transform into cancer cells easily, since cancer arises and progresses through a multi-step process involving deregulation of multiple signaling pathways and cellular functions, and infections caused by a variety of bacterial pathogens may simply contribute one such step. After repeating the experiments for many times, we are confident to demonstrate that the PtpA-expressing BCG does indeed promote proliferation and migration of human lung adenoma A549 cells in vitro and in a mouse xenograft model in our experimental system. We also do not intent to claim that BCG can easily cause cancer in patients receiving BCG vaccination or treatment. Instead, our findings, together with some observations from others, suggest that there is a potential risk in applying BCG treatment for certain patients under certain circumstances, and we should be more cautious and pay closer monitoring while treating those patients with BCG. BCG has been utilized for intravesical immunotherapy for treating cancers such as bladder cancer. However, a significant proportion of the patients fail such immunotherapies. We still do not have clear understanding on the reasons for those failed therapies, though several mechanisms of action of BCG have been proposed. For example, there are multiple evidences of anti-inflammatory Th2 and regulatory T cell functions stimulated by BCG

(Martino et al., 2004; Scott-Browne et al., 2007), and both Th2 and regulatory T cells exhibit pro-tumor effects in cancers (Mougiakakos et al., 2010; Protti et al., 2012). Evasion of protective immune responses by BCG could also contribute to failed immunotherapy. In the present study, we found that infection with PtpA-expressing BCG can promote proliferation and migration of A549 cells, partially through targeting GADD45A. Our results indicated that promotion of cancer cell proliferation by BCG could result in unsuccessful immunotherapy in some patients under certain circumstances.

Furthermore, increasing evidences have been provided to suggest a link between Mtb/BCG and Cancer. For example, it was previously reported that lung carcinogenesis can be induced by chronic tuberculosis infection in a mouse model (Nalbandian et al., 2009), and that BCG can promote the survival of A549 and several other tumor cells from TNF α -induced apoptosis thereby promoting tumorigenesis in xenograft studies (Holla et al., 2014). In addition, Mtb-infected THP-1 cells can induce epithelial mesenchymal transition (EMT) in the lung adenocarcinoma epithelial cell line A549 (Gupta et al., 2016). However, causal links between Mtb/BCG infection and lung cancer in humans have not been fully demonstrated, and there is conflicting evidence concerning a possible association between Mtb-caused pulmonary tuberculosis and subsequent risk of lung cancer (Liang et al., 2009; Shiels et al., 2011).

Taken together, based on the increasing evidence from studies by others and ours, we believe that there is a link between Mtb/BCG and cancer. At the meantime, we also notice that dependent on host immune status, tumor type and stage, as well as the treatment methodology and dosing regimen, BCG might inhibit or promote the tumor progression. The detailed molecular mechanisms underlying the complex relationships among infection, host response and tumor growth warrant more in-depth investigations.

Experimentally they used a cancer model building on a cell line that is not a recognized model for mycobacterial infection and in a multiple of infectivity of 20! which may cause cell lysis and release of carcinogenic matter.

R: The reviewer mentioned that the A549 cell line is not a recognized model for mycobacterial infection and the MOI 20 is high. Actually, it is now well-recognized that alveolar epithelium is among the initial sites of the lung's response against Mtb. Alveolar epithelium is composed of type I and type II cells (Chuquimia et al., 2013). Increasing evidence implicated that alveolar epithelium, particularly type II cells, play an important role in both host cell defense and bacterial dissemination (Xiong et al., 2014; Fine-Coulson et al., 2015; Ryndak et al., 2015). Type II cells could provide a permissive environment for Mtb to replicate and ultimately help bacterial dissemination (Bermudez and Goodman, 1996; Ryndak et al., 2015). The human lung adenoma A549 cell line has been

widely used as a model system for the study of alveolar type II cell function, and it is also a recognized model for mycobacterial infection (Fine et al., 2012; Ma et al., 2014; Zerin et al., 2015; Huang et al., 2015; Ryndak et al., 2015; Adcock et al., 2016; Zheng et al., 2017). We thus believe that this cell line is suitable for investigating the regulatory function of mycobacterial PtpA in tumor cell proliferation. On the other hand, we also noticed that mycobacteria (such as Mtb and BCG) do not enter into epithelial cells as efficiently as into macrophages, and epithelial cells do not provide the optimum conditions for mycobacterial survival and replication. We have tried a variety of MOIs for the infection experiments with A549 cells and found that a MOI of 20 is the optimum MOI for the mouse xenograft model experiments to examine the role of the PtpA-expressing BCG in promoting tumor growth. In our Xenograft tumor model system, we can obtain consistent results without causing cell lysis, and we can also manage to observe stained PtpA in tumors using with a MOI of 20. We also noticed that Fine KL et al. used a MOI of 100 for studying the survival and replication of Mtb within A549 cells (Cell Microbiology, 2012). Actually, except for the nude mice experiments, we used a MOI of 10 for all the other infection experiments with macrophages and A549 cells.

Macrophages infected by BCG or Mtb remain viable and contain the infection. No information was provided as to the fate of BCG with these cell lines or the nude mice. The experiment should be validated before jumping into controversial declarations.

R: Actually, we provided the data for the survival (fate) of BCG in macrophages as shown in Fig. 6d and 8d. Since we used A549 cell line mainly to investigate the regulatory function of mycobacterial PtpA in tumor cell proliferation (instead of for intracellular survival of the pathogen), we thus did not provide the data for the CFU counts of BCG in A549 cells to save space in figures while trying to focus on our main study purpose, though we did obtain those data. To fully address the reviewer's concern on information about the fate of BCG in the cell lines and nude mice, we added the CFUs of BCG in the revised Figures 1, 9 and 10.

Once again, we greatly appreciate the reviewers for having helped us improve our manuscript.